# FUSE: Quantifying Uncertainty in Vision-Language Models by Bayesian Fusing Epistemic and Aleatoric Uncertainty

**Harry Zhang** [1]   **Luca Carlone** [1]

## Abstract

Vision-language models (VLMs) are playing an increasingly important role across multiple domains. In many applications, such as robotics, it is crucial to quantify the uncertainty in the output of these models. We develop FUSE, a probabilistic framework for capturing two complementary sources of uncertainty in vision-language modeling: (i) aleatoric embedding-level uncertainty derived from input data vision-language ambiguity, and (ii) epistemic model-level uncertainty estimated from the semantic response diversity of VLMs. Our approach formulates a Bayesian fusion mechanism that analytically combines these uncertainty sources to produce a scalar measure of uncertainty. This measure can be used to reliably predict the model's output correctness for downstream applications. We demonstrate that our method outperforms baselines and achieves SOTA uncertainty calibration.

## 1. Introduction

Vision-Language Models (VLMs) integrate visual, textual, and sometimes auditory modalities within the same generative framework, enabling open-domain reasoning across diverse input signals. Powered by Transformer-based architectures (Vaswani et al., 2017) and trained on massive web-scale datasets, VLMs now support a wide spectrum of applications, including visual question answering, multimodal retrieval, code generation with image context, and embodied task planning and manipulation (Antol et al., 2015; Lin et al., 2024; O'Neill et al., 2024). Recent advances in scaling laws reveal that increasing model parameters, data diversity, and context length consistently enhances emergent reasoning and cross-modal alignment capabilities (Kaplan et al., 2020; Hoffmann et al., 2022; Yang et al., 2022). How-

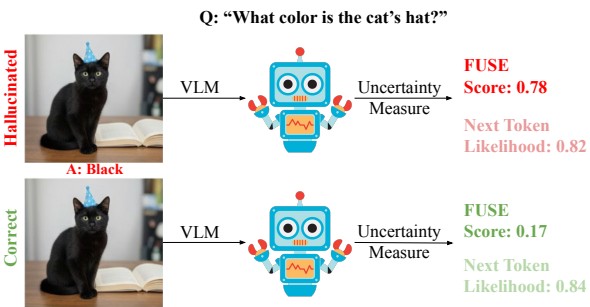

Q: "What color is the cat's hat?"

*Figure 1.* FUSE measures VLM uncertainty via a single uncertainty score. A higher score means more uncertain/hallucinated, and vice versa. Next-token likelihood (NTL) alone cannot distinguish uncertainty as hallucinated samples could have high NTL.

ever, such scaling comes with formidable computational and interpretational costs: as architectures grow, their internal mechanisms become more opaque, and their behavior more difficult to analyze, interpret, and control (Bommasani et al., 2022; Liang et al., 2022).

Despite the fast-paced progress, VLMs remain vulnerable to unreliable or inconsistent outputs, including hallucinated visual descriptions, overconfident textual reasoning, and unstable grounding between modalities, which are especially critical in high-stakes or safety-critical domains such as healthcare, finance, education, and autonomous systems, where users must understand when to trust model predictions (Li et al., 2023; Abbasli et al., 2025). In such contexts, **uncertainty quantification (UQ)** becomes imperative for model reliability: a well-calibrated uncertainty estimate allows the system to understand what it does not know, guiding selective prediction and uncertainty-aware decision making (Abbasli et al., 2025; Zhang & Carlone, 2024). Recent studies show that both unimodal LLMs and multimodal variants often exhibit miscalibration, expressing over-confidence or under-confidence that poorly reflects empirical correctness (Chhikara, 2025; Kapoor et al., 2024; Bai et al., 2024). Conventional methods such as probabilities derived from next-token likelihoods or logit-based confidences fail to capture semantic ambiguity and cross-modal uncertainty. Consequently, the reported confidence of an MLLM's output may not correspond to its true accuracy or

[1]Massachusetts Institute of Technology. Correspondence to: Harry Zhang <harryz@mit.edu>.

grounding quality, thus undermining reliability and safety in deployment.

In this work, we are motivated by the observation that uncertainty in VLMs arises from *two distinct sources*. The first is *aleatoric uncertainty*, which reflects intrinsic ambiguity in the input and its cross-modal semantics (*e.g.*, underspecified questions, visually ambiguous regions, or multiple plausible text–image alignments). The second is *epistemic uncertainty* that comes from the model's generation itself, reflecting the model's variability due to model limitations or data scarcity.

We posit that both sources of uncertainty are indispensable for quantifying uncertainty in VLMs. We propose to model these two sources of uncertainty within a unified Bayesian inference framework. At the input data level, we employ a probabilistic representation using a Gaussian Process (GP), which converts deterministic embeddings from a frozen vision-language encoder such as CLIP (Radford et al., 2021) into distributions that quantify representation uncertainty; this uncertainty is used as the prior for our uncertainty measure. At the model level, we treat the variability of semantics from multiple sampled responses due to limited knowledge, weak grounding, or distribution shift as empirical evidence, summarizing their dispersion via an *evidence* statistic. By connecting these two levels probabilistically, we interpret the GP as a *prior* over latent semantic alignment and the response-level dispersion as a *likelihood* that provides data-driven evidence of uncertainty. We prove the resulting joint Bayesian update yields a closed-form posterior that fuses both sources of uncertainty into a single calibrated measure for downstream applications. Our results indicate that taking advantage of the fusion of two sources of uncertainties, our method achieves state-of-the-art performance across various metrics and datasets.

This leads to **FUSE**: **F**used **U**ncertainty with **S**emantic **E**vidence for VLMs. To summarize, our contributions include:

- Formalization of two fundamental sources of uncertainty in VLMs, providing a conceptual foundation for disentangling and quantifying uncertainty within both model representations and generative outputs.
- A unified probabilistic model that combines the uncertainty sources in a principled Bayesian manner, yielding an interpretable, calibrated posterior uncertainty for each multimodal query.
- Mathematical proof that leads to justification for the Bayesian update used in practice.
- Empirical results on various visual question answering benchmarks that demonstrate the proposed model achieves superior calibration, selective accuracy, and uncertainty-error correlation.

## 2. Related Work

**Multimodal LLMs.** Large language models augmented with multimodal capabilities (*e.g.*, visual-language models) have shown great promise in advancing artificial general intelligence. MLLMs build on large-scale, weakly-supervised alignment of images and text. CLIP (Radford et al., 2021) popularized dual-encoder contrastive learning over 400M image-text pairs, enabling zero-shot transfer for classification and retrieval. Closely-related works such as ALIGN and BLIP (Jia et al., 2021; Li et al., 2022) show that scale can compensate for noise, improving zero-shot retrieval and classification. Flamingo and variants introduce a few-shot visual language model that conditions a frozen LLM on interleaved image-text context via cross-attention perceiver resamplers (Alayrac et al., 2022; Awadalla et al., 2023). MiniGPT-4 and follow-ups (Zhu et al., 2023; Wei et al., 2023) show that aligning visual features to a strong LLM with a minimal projector yields rich multimodal behaviors after alignment data. LLaVA connects a pretrained CLIP vision encoder to an LLM and is instruction-tuned on synthetic multimodal dialogs, becoming a widely used open baseline (Liu et al., 2023b; Li et al., 2023; Liu et al., 2024a;b; Li et al., 2024). Many subsequent open-source MLLMs follow the same paradigm, substituting improved CLIP derivatives such as (Liang et al., 2023; Sun et al., 2023a) for higher-resolution features. Qwen-VL families integrate grounding, OCR, and multilingual chat into instruction-tuned MLLMs at scale (Bai et al., 2023; Wang et al., 2024b). Describe Anything (Lian et al., 2025) uses CLIP-based confidence filtering for localized captioning tasks. This reliance on frozen CLIP encoders thus combines efficient reuse of large-scale visual representations with flexible instruction-tuned language backbones. Inspired by this trend, we tackle the first source of uncertainty, the data itself, at the embedding level, based on the disagreement between visual and textual inputs, which is given "for free" by CLIP-based methods. However, CLIP and alike produce deterministic embeddings that do not lend themselves well to uncertainty quantification. We adapt the embeddings probabilistically using a latent Gaussian Process.

**Uncertainty Quantification.** Before the LLM era, extensive research in classical deep learning established the foundational paradigms for uncertainty quantification. Gal & Ghahramani (2016) propose Monte Carlo Dropout, interpreting dropout at inference time as approximate Bayesian inference to capture epistemic uncertainty. Lakshminarayanan et al. (2017) introduced Deep Ensembles, averaging predictions from independently trained networks to obtain robust predictive variance estimates. Bayesian neural networks (Blundell et al., 2015) model posterior distributions over network weights but suffered from scalability challenges, inspiring subsequent work on variational inference

and stochastic gradient MCMC methods (Neal, 2012; Papamarkou et al., 2022). To quantify data-driven (aleatoric) uncertainty, Kendall & Gal (2017) formulate multi-task loss functions combining homoscedastic and heteroscedastic terms. Calibration-oriented works such as Guo et al. (2017) demonstrated that post-hoc transformations can align model confidence with empirical accuracy across deep classifiers. Further developments such as Evidential Deep Learning (Sensoy et al., 2018) unify aleatoric and epistemic components under a single Dirichlet predictive distribution, while conformal prediction frameworks (Shafer & Vovk, 2008; Angelopoulos & Bates, 2021; Barber et al., 2023) provide distribution-free uncertainty sets with finite-sample coverage guarantees. While classical UQ methods in deep learning provide principled uncertainty, they do not resolve uncertainty for multimodal LLMs. In MLLMs token-level likelihood or entropy is often weakly correlated with semantic correctness or visual grounding (Chou et al., 2025). The "confidence" being calibrated is not aligned with the event of interest (*i.e.*, correctness) (Lau et al., 2025; Chen et al., 2024). These gaps motivate uncertainty statistics that (i) are lightweight for MLLMs, and (ii) quantify uncertainty at the grounding level rather than purely at the token level.

In MLLMs, the synthesis of text and visual modalities introduces compounding sources of uncertainty. Each sub-process of MLLM inference contributes its own form of uncertainty (Bai et al., 2024). While recent works attempt to mitigate hallucinations through targeted training or data strategies (Liu et al., 2023a; Yu et al., 2024; Wang et al., 2024a; Yue et al., 2024), architectural adjustments (Tong et al., 2024; Zhai et al., 2023), or modified learning procedures (Jiang et al., 2024), such errors are difficult to detect or eliminate entirely in real-world, noisy multimodal data. A complementary direction focuses on error and uncertainty estimation at inference time. Input-level approaches, which measure data-level uncertainty, attempt to assess uncertainty by measuring the ambiguity of multimodal embeddings themselves (Tran et al., 2022; Upadhyay et al., 2023; Venkataramanan et al., 2025), providing indicators of representation reliability but not capturing the uncertainty of generated outputs. A growing body of work estimates output-level uncertainty by evaluating the semantic consistency of stochastic generations. These *sampling-based* approaches treat a model's variability as signal: when multiple responses diverge semantically, the system is likely uncertain. This family includes entropy-based metrics (Nikitin et al., 2024; Farquhar et al., 2024), weighted self-consistency scores (Manakul et al., 2023; Lin et al., 2023), and geometric or volumetric statistics in embedding space (Lau et al., 2025; Chen et al., 2024). Another line of work applies distribution-free conformal prediction techniques to generative models (Quach et al., 2023; Su et al., 2024), ensuring reliable coverage guarantees for selective prediction. Furthermore, batch-level calibration methods such as Batch Calibration (Zhou et al., 2023) mitigate prompt- or context-specific biases, making confidence scores more comparable across inference batches. Other studies evaluate reliability through response perturbation and verification. For example, Zhang et al. (2024); Sun et al. (2023b); Khan & Fu (2024); Liu et al. (2023a) employ external verifiers or response perturbations to assess the consistency of MLLM outputs as an indicator of uncertainty. However, existing MLLM UQ methods largely operate at the generation level, often overlooking uncertainty embedded in the model's latent representation itself. Current approaches often treat uncertainty scores as heuristic rather than principled probabilistic quantities. In this work, we aim to bridge this gap by proposing a joint Bayesian formulation that integrates input representation-level priors derived from latent probabilistic embedding, with generation-level evidence modeled through semantic response diversity.

## 3. Problem Statement

We consider a vision-language model (VLM) that generates textual responses conditioned on visual and textual inputs. Let an input query be denoted by $q = (\mathbf{I}, \mathbf{T})$, where $\mathbf{I}$ represents the visual input (*e.g.*, image) and $\mathbf{T}$ the textual prompt. The model produces a response $y$ sampled from its conditional distribution $y \sim p_\theta(y \mid q)$, where $\theta$ denotes fixed VLM model parameters.

A typical VLM architecture encodes the input modalities via a frozen vision-language encoder, which gets fed into the downstream LLM backbone. We assume access to the model's frozen vision-language encoder (*e.g.*, CLIP), providing deterministic embeddings for the input modalities $\mathbf{z}_I = f_I(\mathbf{I}), \mathbf{z}_T = f_T(\mathbf{T}) \in \mathbb{R}^D$. We then denote by $\mathbf{r}_i \in \mathbb{R}^d$ the hidden-state representation of the $i$-th sampled response $y_i$ for $i = 1, \ldots, n$, produced by LLM backbone.

We distinguish two complementary forms of uncertainty. *Aleatoric uncertainty* arises from inherent ambiguity or noise in the input data or task context. In this work, it is represented by the mean $\boldsymbol{\mu}_I$ and the covariance $\boldsymbol{\Sigma}_I$ of the visual input embedding, and $\boldsymbol{\mu}_T$ and $\boldsymbol{\Sigma}_T$ of the textual input embedding, reflecting ambiguity in multimodal alignment (*e.g.*, multiple valid image-text correspondences). *Epistemic uncertainty*, on the other hand, arises from variability in the model's predictions for a fixed input due to weak knowledge or distribution shift. We capture this via the dispersion of sampled response embeddings $\{\mathbf{r}_i\}_{i=1}^{n}$ from last hidden layer vector of the last response token, summarized through a *semantic evidence* statistic $\tilde{V}$ from the Gram matrix of responses, which quantifies the geometric diversity of responses in the model's semantic space.

Given $(\boldsymbol{\mu}_I, \boldsymbol{\Sigma}_I, \boldsymbol{\mu}_T, \boldsymbol{\Sigma}_T)$ from the probabilistically adapted

encoder and $\tilde{V}$ from stochastic decoding, our goal is to estimate a scalar *score*: $u^\star = f(\boldsymbol{\mu}_I, \boldsymbol{\Sigma}_I, \boldsymbol{\mu}_T, \boldsymbol{\Sigma}_T, \tilde{V})$, that faithfully reflects the model's confidence in its output (*i.e.*, the more uncertain the model is, the less likely its output is correct), satisfying $\mathbb{P}[\text{correct} \mid u^\star] \approx 1 - u^\star$, and enabling interpretability of uncertainty values to support downstream decisions such as selective answering.

## 4. Methods

Given a multimodal query (*e.g.*, an image-text pair), FUSE produces two complementary uncertainty signals:

1. A *data representation prior* $u \sim \mathcal{N}(m_0, \sigma_0^2)$ that reflects embedding-level uncertainty from Definition 4.1;
2. A *semantic evidence* $\tilde{V} \mid u$ summarizing stochastic diversity across $n$ responses (Definition 4.2, Definition 4.7).

Combining these via the linear-Gaussian likelihood (Theorem 4.4-Theorem 4.8) yields a conjugate posterior distribution for $u$ (Theorem 4.9), from which we can derive a scalar uncertainty score $u^\star$ — a single measure to quantify the VLM's uncertainty.

### 4.1. Aleatoric Uncertainty via VLM Embeddings Prior

Let a frozen vision-language model (*e.g.*, CLIP) provide deterministic $D$-dimensional embeddings $\boldsymbol{z}_I = f_I(\mathbf{I})$ and $\boldsymbol{z}_T = f_T(\mathbf{T})$ for each paired input $(\mathbf{I}, \mathbf{T})$. Our first objective is to generalize deterministic embeddings to probabilistic distributions that can capture the aleatoric uncertainty. We generalize these point embeddings to uncertainty-aware distributions by introducing a shared latent variable $\boldsymbol{x} \in \mathbb{R}^Q$ with $Q \ll D$, where each pair $(\mathbf{I}, \mathbf{T})$ is associated with the same latent point $\boldsymbol{x}$, representing the "true" meaning of the data in latent space.

We learn two Gaussian Process (GP) decoders that map $\boldsymbol{x}$ to the embedding spaces:

$$\boldsymbol{z}_\mathcal{M} = G_\mathcal{M}(\boldsymbol{x}) + \boldsymbol{\varepsilon}_\mathcal{M}, \quad \mathcal{M} \in \{\text{I}, \text{T}\}, \tag{1}$$

with noise $\boldsymbol{\varepsilon}_\mathcal{M}$. Each $G_\mathcal{M}$ is $D$-dimensional. For each output dimension $k \in \{1, \ldots, D\}$, we place a scalar GP prior: $g_k^{(\mathcal{M})}(\cdot) \sim \mathcal{GP}\big(m_\mathcal{M}, k_\mathcal{M}(\cdot, \cdot)\big)$, $k_\mathcal{M}(\boldsymbol{x}, \boldsymbol{x}') = \exp\big[-\frac{\|\boldsymbol{x}-\boldsymbol{x}'\|_2^2}{2\ell_\mathcal{M}^2}\big]$. A full GP latent model scales cubically in the dataset size $N$. We adopt a sparse variational GPLVM following (Lawrence, 2003; Venkataramanan et al., 2025) using $N_u \ll N$ inducing points, which act as a small set of "anchors" that summarize each GP in latent space: once we know the GP function values at these anchors, the GP can interpolate to all training and test latent locations. For each modality $\mathcal{M}$ and output dimension $k$, we choose inducing locations $\{\boldsymbol{u}_j^{(\mathcal{M})}\}_{j=1}^{N_u} \subset \mathbb{R}^Q$ and define inducing values $\boldsymbol{i}_k^{(\mathcal{M})} = \big[g_k^{(\mathcal{M})}(\boldsymbol{u}_1^{(\mathcal{M})}), \ldots, g_k^{(\mathcal{M})}(\boldsymbol{u}_{N_u}^{(\mathcal{M})})\big]^\top \in$

$\mathbb{R}^{N_u}$. We posit a Gaussian variational posterior over inducing values: $q(\boldsymbol{i}_k^{(\mathcal{M})}) = \mathcal{N}(\boldsymbol{\mu}_k^{(\mathcal{M})}, \boldsymbol{S}_k^{(\mathcal{M})})$, and optimize its parameters $\boldsymbol{\mu}_k^{(\mathcal{M})}, \boldsymbol{S}_k^{(\mathcal{M})}$ by maximizing an ELBO. Let $\boldsymbol{g}_k^{(\mathcal{M})} = [g_k^{(\mathcal{M})}(\boldsymbol{x}_1), \ldots, g_k^{(\mathcal{M})}(\boldsymbol{x}_N)]^\top$ collect latent function values at the training latent points. The ELBO for modality $\mathcal{M}$ and dimension $k$ is $\mathcal{L}_{\text{ELBO}}^{(\mathcal{M}, k)} = \mathbb{E}_q\big[\log p(\boldsymbol{z}_{:,k}^{(\mathcal{M})} \mid \boldsymbol{g}_k^{(\mathcal{M})})\big] - \text{KL}\big(q(\boldsymbol{i}_k^{(\mathcal{M})}) \| p(\boldsymbol{i}_k^{(\mathcal{M})})\big)$, where $p(\boldsymbol{i}_k^{(\mathcal{M})})$ is the GP prior at the inducing locations. Summing over dimensions and modalities (Venkataramanan et al., 2025) yields the *embedding reconstruction loss* $\mathcal{L}_{\text{emb}} = -\sum_{\mathcal{M} \in \{\text{I}, \text{T}\}} \sum_{k=1}^{D} \mathcal{L}_{\text{ELBO}}^{(\mathcal{M}, k)}$. Since both modalities share the same $\boldsymbol{x}$ for each pair, we align the two predictive embedding distributions at $\boldsymbol{x}$. Let $\mathcal{F}_\text{I}$ and $\mathcal{F}_\text{T}$ denote the GP predictive Gaussians for the two modalities. We penalize the symmetrized KL divergence $\mathcal{L}_{\text{KL}} = \frac{1}{2}\big[\text{KL}\big(\mathcal{F}_\text{I} \| \mathcal{F}_\text{T}\big) + \text{KL}\big(\mathcal{F}_\text{T} \| \mathcal{F}_\text{I}\big)\big]$, which has a closed form in terms of predictive means and covariances. The final objective is then

$$\mathcal{L}_{\text{total}} = \lambda_1 \mathcal{L}_{\text{emb}} + \lambda_2 \mathcal{L}_{\text{KL}}, \qquad \lambda_1, \lambda_2 > 0. \tag{2}$$

Optimizing $\mathcal{L}_{\text{total}}$ learns: (i) latent points $\{\boldsymbol{x}_n\}_{n=1}^N$ shared by both modalities, (ii) inducing locations $\{\boldsymbol{u}_j^{(\mathcal{M})}\}$, (iii) variational parameters $\{\boldsymbol{\mu}_k^{(\mathcal{M})}, \boldsymbol{S}_k^{(\mathcal{M})}\}$, and (iv) GP hyperparameters and noise, producing probabilistic embeddings whose predictive covariances quantify uncertainty.

Given a new deterministic embedding $\boldsymbol{z}^\star$ (image or text), we infer its latent representation $\boldsymbol{x}^\star$ by solving $\boldsymbol{x}^\star = \arg\max_{\boldsymbol{x}} p(\boldsymbol{x} \mid \boldsymbol{z}^\star; \hat{\theta})$, where $\hat{\theta}$ denotes the learned GP hyperparameters and inducing variables. We then obtain a probabilistic embedding from the predicted GPs, where the mean and covariance $\mathcal{N}(\hat{\boldsymbol{\mu}}^\star, \hat{\boldsymbol{\Sigma}}^\star)$ provide the *probabilistic embedding* and its corresponding uncertainty estimate.

We now propose a hierarchical Bayesian coupling: the learned GPs provide representation-level uncertainty that induces a prior over a variable $u$, which will be refined using uncertainty extracted from the VLM's sampled responses. We start by transforming the GPs into a scalar uncertainty measure that we call the "data representation prior".

**Definition 4.1** (Data Representation Prior). We introduce a scalar uncertainty variable $u \in \mathbb{R}$ that summarizes the overall ambiguity of the query at the representation level.

Let the GP predictive distribution for each modality be $\mathcal{F}_\text{I} = \mathcal{N}(\boldsymbol{\mu}_\text{I}, \boldsymbol{\Sigma}_\text{I})$ and $\mathcal{F}_\text{T} = \mathcal{N}(\boldsymbol{\mu}_\text{T}, \boldsymbol{\Sigma}_\text{T})$. We summarize these modality-specific uncertainties into a scalar statistic $s = g(\boldsymbol{\mu}_I, \boldsymbol{\Sigma}_I, \boldsymbol{\mu}_T, \boldsymbol{\Sigma}_T)$, where $g(\cdot)$ is a fixed aggregation function chosen to be monotone with respect to the dispersion. We model the prior distribution of $u$ as a Gaussian,

$$u \sim \mathcal{N}(m_0, \sigma_0^2), \qquad m_0 = \alpha_0 + \beta_0 s, \tag{3}$$

The parameters $(\alpha_0, \beta_0, \sigma_0^2)$ are calibrated on a held-out calibration set. The procedure is shown in Appendix F.

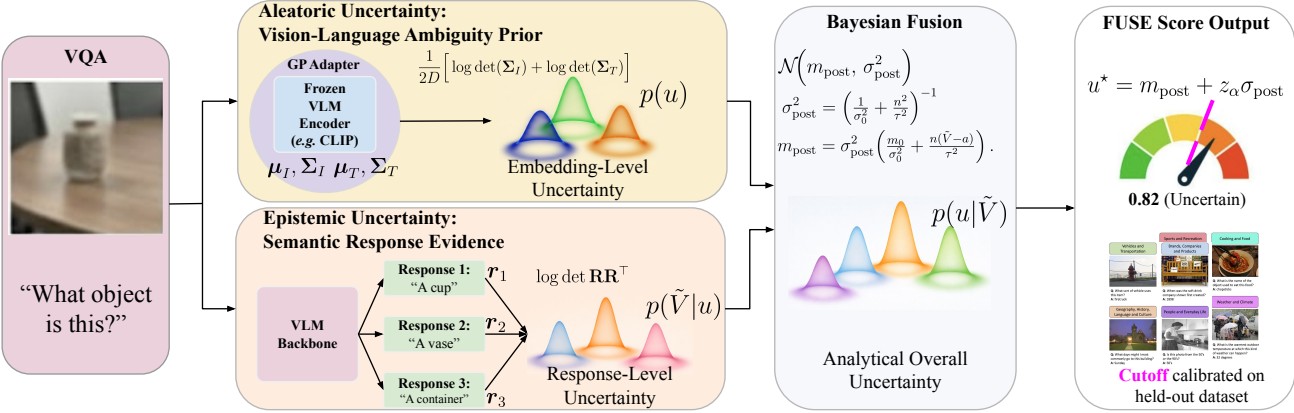

*Figure 2.* FUSE Overview. We extract aleatoric uncertainty from VLM's frozen encoder and use it as the prior. We extract epistemic uncertainty from the model's response and use it as the evidence. We then use Bayesian fusion to obtain an analytical posterior.

The form of $m_0$ can be understood as a first-order approximation relating encoder-level uncertainty to latent task uncertainty and we choose it for mathematical convenience and simplicity. The exact form of $s = g(\boldsymbol{\mu}_I, \boldsymbol{\Sigma}_I, \boldsymbol{\mu}_T, \boldsymbol{\Sigma}_T)$ is a design choice, but it should measure the *dispersion* of the input representation. We choose to use a variance-based metric $s = \frac{1}{2D}\left[\log\det(\boldsymbol{\Sigma}_I) + \log\det(\boldsymbol{\Sigma}_T)\right]$. Hence, this prior captures the uncertainty contributed by the input itself. The above procedure is independent of downstream VLM inference. Therefore, our decomposition separates aleatoric uncertainty from epistemic uncertainty.[1]

## 4.2. Epistemic Uncertainty via Response Semantics Evidence

Next, we build a likelihood from epistemic uncertainty by converting response diversity into a scalar evidence statistic. We (i) sample $n$ responses, (ii) embed each response as $\mathbf{r}_i$, (iii) aggregate them into a scatter matrix, and (iv) use the $\log\det$ of the matrix as evidence. Under mild assumptions, we show this likelihood can be fused with the prior introduced above using a closed-form Bayesian update.

**Definition 4.2** (Model Response). Conditional on the uncertainty level $u$, we model each sampled $d$-dimensional response embedding as

$$\mathbf{r}_i \mid u \stackrel{\text{i.i.d.}}{\sim} \mathcal{N}\left(0, e^u\boldsymbol{\Sigma}_r\right), \qquad i = 1, \ldots, n, \quad (4)$$

where $\boldsymbol{\Sigma}_r \in \mathbb{R}^{d \times d}$ is a fixed positive-definite *shape matrix* describing the base geometry of the response space.

This assumption implies that higher $u$ corresponds to greater variability among responses. We follow (Lau et al., 2025) and normalize the response embeddings.[2] Next, we collect the model responses using a Gram matrix to form the semantic scatter and evidence for inference.

[1]Further justification is provided in Appendix A.

[2]More details are provided in Appendix G.

**Definition 4.3** (Semantic Scatter and Evidence). Given uncertainty level $u$ with whitened $\{\mathbf{r}_i\}_{i=1}^n$, stack rows into $\boldsymbol{R} = [\boldsymbol{r}_1, \cdots, \boldsymbol{r}_n]^\top \in \mathbb{R}^{n \times d}$. Let the semantic scatter matrix be $\mathbf{S} = \boldsymbol{R}\boldsymbol{R}^\top$. We define the semantic evidence $\tilde{V} := \log\det \mathbf{S}$.

The scatter $\mathbf{S} = \boldsymbol{R}\boldsymbol{R}^\top$ summarizes how response embeddings fill the semantic space. The determinant is proportional to the squared volume of the ellipsoid spanned by the response; hence $\tilde{V}$ measures semantic dispersion. To build the likelihood, we first analyze the *unweighted* case, where all sampled responses contribute equally. The following result establishes the key structure that the $\log\det$ of the response scatter matrix is exactly linear in $u$.

**Theorem 4.4** (Linearity of Semantic Evidence in $u$). *Given the unweighted semantic scatter* $\mathbf{S}$ *and its corresponding semantic evidence* $\tilde{V}$ *in Definition 4.3, we then have:*

$$\tilde{V} = n\,u + \log\det\boldsymbol{\Lambda}, \quad (5)$$

*where* $\boldsymbol{\Lambda} \sim \mathsf{Wishart}_n(\boldsymbol{I}, d)$ *is a $n$-dimensional Wishart random matrix.*

Hence, $\tilde{V}$ depends *exactly linearly* on $u$ with slope $n$, the dimension of response embeddings. Increasing $u$ scales each response embedding by $\sqrt{e^u}$, which scales the scatter matrix by $e^u$. Intuitively, since $\det(e^u\mathbf{S}) = e^{nu}\det(\mathbf{S})$, taking logs yields an additive shift $nu$.

Under this construction, the semantic evidence's sampling distribution becomes a closed-form Gaussian.

**Theorem 4.5** (Gaussian Likelihood of Semantic Evidence). *The semantic evidence* $\tilde{V}$ *defined in Definition 4.3, for given $d$ and $n$, follows* $\tilde{V} \mid u \sim \mathcal{N}\left(a(n, d) + n\,u,\ v(n, d)\right)$, *where $a(n, d)$ and $v(n, d)$ are analytical functions of $n$ and $d$ defined in Appendix C.*

Consequently, the evidence statistic $\tilde{V} = \log\det \mathbf{S}$ admits a simple *linear-Gaussian* form: its mean shifts linearly with

the latent uncertainty level $u$, and its variance is determined explicitly by the finite-sample Wishart moments.

**From Unweighted to Weighted Responses.** In practice, sampled responses often have heterogeneous reliability: depending on the query, the VLM may generate responses with various levels of coherence due to the model's stochasticity. We therefore introduce *uncertainty weights* that modulate each response's contribution to the overall evidence.

**Definition 4.6** (Uncertainty Weights). For each sampled response $y_i$, let $p_i$ denote the average per-token log-probability assigned by the model. We define the corresponding weight as $w_i = \exp(2\alpha(1 - p_i))$, where $\alpha$ is a hyperparameter. Responses with lower confidence ($p_i$) thus receive higher weights, increasing their influence in the semantic scatter.

We then define the semantic scatter and evidence with the uncertainty weights.

**Definition 4.7** (Weighted Semantic Scatter and Evidence). Given weights $\{w_i\}_{i=1}^n$, collect responses and weights into $\boldsymbol{R} = [\boldsymbol{r}_1, \cdots, \boldsymbol{r}_n]^\top$ and $\boldsymbol{W} = \mathrm{diag}(w_1, \cdots, w_n)$, we define the weighted semantic scatter and its associated semantic evidence as $\mathbf{S}_w := \boldsymbol{W}^{1/2}\boldsymbol{R}\boldsymbol{R}^\top\boldsymbol{W}^{1/2}$, $\tilde{V} = \log\det\mathbf{S}_w$. The unweighted case corresponds to $w_i \equiv 1$, where $\mathbf{S}_w = \mathbf{S}$ and $\tilde{V} = \log\det\mathbf{S}$.

Similar to Theorem 4.4, we obtain a weighted version of the likelihood of the semantic evidence, which follows the same linear dependence on $u$.

**Theorem 4.8** (Weighted Semantic Evidence Likelihood). *Assume the weights $\{w_i\}$ are bounded away from $0$ and $\infty$, Then for fixed $d$ and $n$, we have*

$$\tilde{V} = nu + a_w(\nu_{\mathrm{eff}}, d, n, \bar{w}) + \varepsilon_w, \qquad (6)$$

*where $\varepsilon_w \sim \mathcal{N}\big(0, v(d, n)\big)$. Here, $\bar{w} = \frac{1}{n}\sum_{i=1}^n w_i$, $\nu_{\mathrm{eff}} = \frac{(\sum_i w_i)^2}{\sum_i w_i^2}$, where $a_w(n, d)$ and $v(n, d)$ are analytical functions of $n, d$ defined in Appendix D.*

Here, $a_w(\cdot)$ and $v(\cdot)$ are weighted analogues of $a, v$ from the unweighted case. Importantly, the dependence on $u$ retains the *same slope* $n$. Weighting effectively reduces the number of independent samples contributing to the scatter and the *effective degrees of freedom* $\nu_{\mathrm{eff}}$ captures this reduction.

**Linear-Gaussian Evidence Model.** Theorems 4.4-4.8 yield a unified, tractable evidence model for $\tilde{V}$:

$$\tilde{V} \mid u \sim \mathcal{N}\big(a + nu, \tau^2\big), \qquad (7)$$

where $(a, \tau^2)$ are given by the closed-form formulas (possibly with $\nu_{\mathrm{eff}}, \bar{w}$ as functions of $n, d$) (see Appendix C and Appendix D.) This Gaussian distribution provides (i) a canonical scale for comparing uncertainty across queries,

since $\tau^2$ depends explicitly on $(n, d)$, and (ii) a closed-form Bayesian update when combined with the encoder-level prior. This yields a training-free posterior $p(u \mid \tilde{V})$ and enables calibrated selective prediction via a single threshold.

### 4.3. Bayesian Fusion of the Uncertainty Sources

With both the prior using aleatoric uncertainty from input embeddings and the likelihood using epistemic uncertainty from semantic evidence, we can now fuse the two sources of uncertainty via Bayesian inference, yielding a single fused uncertainty measure for downstream applications.

**Theorem 4.9** (Closed-Form Posterior of Uncertainty). *Combining the data-representation prior (Definition 4.1) with the linear-Gaussian evidence model gives*

$$u \mid \tilde{V} \sim \mathcal{N}\big(m_{\mathrm{post}}, \sigma_{\mathrm{post}}^2\big), \qquad (8)$$

*where* $\sigma_{\mathrm{post}}^2 = \left(\frac{1}{\sigma_0^2} + \frac{n^2}{\tau^2}\right)^{-1}, m_{\mathrm{post}} = \sigma_{\mathrm{post}}^2\left(\frac{m_0}{\sigma_0^2} + \frac{n(\tilde{V}-a)}{\tau^2}\right)$, *with $a, \tau^2$ given in Equation (7).*

Using the posterior uncertainty mean and variance, we design the following single uncertainty statistic.

**Definition 4.10** (Fused Uncertainty Statistic). Given posterior uncertainty parameters $m_{\mathrm{post}}$ and $\sigma_{\mathrm{post}}^2$, we define the single Fused Uncertainty Statistic $u^\star$ as the following z-score-based metric: $u^\star = m_{\mathrm{post}} + z_\alpha\sigma_{\mathrm{post}}$, where $z_\alpha$ is a hyperparameter, yielding a $z_\alpha$-std bound on the uncertainty.

### 4.4. Practical Calibration and Hyperparameters

Before deployment, several quantities must be calibrated on a held-out development set with groundtruth VQA labels:

- The prior parameters $(\alpha_0, \beta_0, \sigma_0^2)$ from Definition 4.1, fitted by regressing empirical error rates on the compressed covariance statistic $s$ using OLS shown in Appendix F.
- The scaling constant $\alpha$ for the uncertainty weights in Definition 4.6, tuned to balance incoherent samples.
- The decision function of the Fused Uncertainty Statistic is calibrated via a single global threshold $\theta^\star$ on the calibration set to control the risk on a size-$N$ held-out calibration set over each calibration query $q$: define the *conditional risk* at threshold $\theta$ as $R(\theta) = \frac{\sum_{q=1}^N \mathbf{1}\{u_q^\star \leq \theta, \hat{y}_q \neq y_q\}}{\sum_{q=1}^N \mathbf{1}\{u_q^\star \leq \theta\}}$. Given a user-specified maximum tolerable error rate $r \in [0, 1]$, the optimal global threshold is chosen as $\theta^\star = \sup_\theta\{R(\theta) \leq r\}$.

At deployment, the system *answers* if $u^\star \leq \theta^\star$ and *abstains* otherwise. This ensures the empirical risk among answered instances does not exceed $r$ on the calibration set, while maximizing coverage. Once calibrated, these parameters

remain fixed and reused across test queries.[3]

# 5. Experiments

In our experiments, we use Llava-v1.5-13b (Liu et al., 2023b), with CLIP-ViT-L-336px (Radford et al., 2021) as its vision backbone. To calculate the prior of $u$, we thus train wrappers for both the vision and text towers of CLIP-ViT-L-336px to adapt them probabilistically with latent GPs. Following (Kuhn et al., 2023; Lau et al., 2025), we evaluate FUSE's performance on three distinct visual question-answering (VQA) benchmark datasets: VQAv2 (Goyal et al., 2017), OKVQA (Marino et al., 2019), and AdVQA (Li et al., 2021). We present only quantitative results here and defer qualitative analyses to Appendix L.

## 5.1. Baselines

We compare FUSE against a variety of uncertainty-quantification baselines, some of which are LLM-specific, but we adapt them to accommodate multi-modality.

**UMPIRE** (Lau et al., 2025): a training-free UQ method for VLMs. It estimates uncertainty from the semantic diversity of multiple sampled responses and weights it by the model's incoherence score. **Neighborhood Consistency** (Khan & Fu, 2024): examines the consistency of the model responses over rephrased questions generated by a small proxy VQG model. **LN-Entropy** (Malinin & Gales, 2020): normalizes the joint log-probability of each sequence by dividing it by the sequence length sampled via multinomial sampling. **Semantic Entropy** (Kuhn et al., 2023): models the uncertainty over different meanings by clustering the generated sequences by (He et al., 2020) and measuring the clusters' entropies. **EigenScore** (Chen et al., 2024): computes the log determinant of the covariance matrix of response embeddings via SVD. **LoFreeCP** (Su et al., 2024): uses conformal prediction and formulates nonconformity measures using both coarse sample frequency and fine-grained semantic similarity without needing to access raw logits. **CLM** (Quach et al., 2023): a logit-based CP method, which uses the general risk control framework FWER (Angelopoulos et al., 2025). We use nonconformity measures from the CP methods as their uncertainty statistics.

## 5.2. Uncertainty as Correctness Prediction

We evaluate whether the fused uncertainty posterior produced by FUSE is a reliable proxy for response correctness.

**Metrics.** A good uncertainty score should satisfy: *discriminative power* (separating correct vs. incorrect answers) and *calibration* (matching uncertainty to empirical error rates).

---

[3]A justification for our risk-control decision rule is provided in Appendix H.

**(1) Discrimination: does $u^\star$ rank incorrect answers as more uncertain?** Let $u^\star_{correct}$ and $u^\star_{incorrect}$ denote the uncertainty scores of correct and incorrect responses, respectively. Our target quantity is the pairwise ranking probability $\mathbb{P}(u^\star_{correct} < u^\star_{incorrect})$, *i.e.*, how often a randomly chosen correct response receives a lower uncertainty score than a randomly chosen incorrect response. Following (Lau et al., 2025; Kuhn et al., 2023), we summarize this ranking quality with the **AUROC** of a binary classifier that uses $u^\star$ to predict correctness (lower $u^\star$ indicates higher confidence / more likely correct). In addition, for a chosen operating threshold on $u^\star$ we report the corresponding **true positive rate (TPR)** given multiple **false positive rate (FPR)** requirements.

**(2) Calibration: does predicted uncertainty match empirical error?** We test whether the (*normalized*) uncertainty score is calibrated in the sense that $\mathbb{P}(\text{correct} \mid u^\star) \approx 1 - u^\star$. We follow the standard binning-based calibration protocol of (Guo et al., 2017): we sort and bin instances by $u^\star$ and compute per-bin average uncertainty and empirical accuracy. We report: **Calibration correlation**, the correlation between binned mean uncertainty and binned empirical error. We provide both **Pearson** and **Spearman** correlations. **Expected Calibration Error (ECE)**, which measures the (weighted) average absolute gap between binned predicted confidence and binned accuracy.

**Results.** As shown in Table 1, FUSE consistently outperforms all baselines on both discrimination (AUROC/TPR/FPR) and calibration (correlations/ECE), including multimodal-targeted methods such as UMPIRE and Neighborhood Consistency. On the more challenging datasets (OKVQA and AdVQA), where model outputs exhibit higher diversity and incoherence, FUSE remains robust and provides substantially better correctness prediction.

## 5.3. Selective Answering

We also evaluate whether FUSE can improve *selective answering—i.e.*, abstaining on uncertain queries and answering only when the model is confident.

**Metrics.** Let $u^\star_q$ be the uncertainty score for query $q$ (lower $u^\star$ indicates higher confidence). For any threshold $\theta$, the selective policy answers iff $u^\star_q \leq \theta$ and abstains otherwise. Varying $\theta$ induces a trade-off between *coverage* (fraction answered) and *accuracy* on the answered subset. Following (Farquhar et al., 2024; Lau et al., 2025; Hüllermeier & Waegeman, 2021), we plot the Rejection–Accuracy curve: we sort queries by $u^\star$ and progressively reject the most uncertain fraction; at each rejection rate $r \in [0, 1]$ (equivalently, coverage $1 - r$), we compute the accuracy on the remaining answered queries. We summarize the overall benefit of uncertainty-based abstention with the **Area Under the Rejection–Accuracy Curve (AURAC)**, which aggregates performance across all thresholds (higher is better).

FUSE: Quantifying Uncertainty in Vision-Language Models by Bayesian Fusing Epistemic and Aleatoric Uncertainty

| | AUROC (↑) | | | TPR @ 0.1 FPR (↑) | | | TPR @ 0.05 FPR (↑) | | | TPR @ 0.01 FPR (↑) | | | Pearson Corr. (↑) | | | Spearman Corr. (↑) | | | ECE (↓) | | | AURAC (↑) | | |
|---|---|---|---|---|---|---|---|---|---|---|---|---|---|---|---|---|---|---|---|---|---|---|---|---|
| Method | VQA | OK | AdV | VQA | OK | AdV | VQA | OK | AdV | VQA | OK | AdV | VQA | OK | AdV | VQA | OK | AdV | VQA | OK | AdV | VQA | OK | AdV |
| UMPIRE | 0.862 | 0.750 | 0.773 | 0.629 | 0.368 | 0.477 | 0.412 | 0.210 | 0.272 | 0.230 | 0.091 | 0.185 | 0.933 | 0.945 | 0.960 | 0.890 | 0.934 | 0.927 | 0.042 | 0.041 | 0.044 | 0.916 | 0.761 | 0.761 |
| Neighborhood | 0.761 | 0.512 | 0.660 | 0.362 | 0.095 | 0.189 | 0.170 | 0.031 | 0.069 | 0.049 | 0.008 | 0.019 | 0.789 | 0.577 | 0.543 | 0.712 | 0.580 | 0.703 | 0.331 | 0.502 | 0.353 | 0.886 | 0.629 | 0.683 |
| LN-Entropy | 0.763 | 0.698 | 0.639 | 0.282 | 0.244 | 0.168 | 0.112 | 0.113 | 0.106 | 0.057 | 0.030 | 0.066 | 0.563 | 0.856 | 0.913 | 0.566 | 0.699 | 0.900 | 0.055 | 0.057 | 0.072 | 0.899 | 0.741 | 0.705 |
| Semantic Entropy | 0.840 | 0.712 | 0.757 | 0.574 | 0.321 | 0.420 | 0.333 | 0.145 | 0.264 | 0.177 | 0.068 | 0.124 | 0.909 | 0.444 | 0.799 | 0.897 | 0.532 | 0.781 | 0.056 | 0.189 | 0.182 | 0.902 | 0.734 | 0.742 |
| EigenScore | 0.850 | 0.741 | 0.766 | 0.601 | 0.340 | 0.466 | 0.420 | 0.195 | 0.212 | 0.215 | 0.075 | 0.172 | 0.922 | 0.790 | 0.872 | 0.883 | 0.842 | 0.811 | 0.056 | 0.190 | 0.211 | 0.913 | 0.753 | 0.753 |
| LoFreeCP | 0.721 | 0.670 | 0.682 | 0.599 | 0.323 | 0.305 | 0.392 | 0.190 | 0.298 | 0.200 | 0.092 | 0.163 | 0.811 | 0.484 | 0.733 | 0.840 | 0.621 | 0.788 | 0.074 | 0.210 | 0.223 | 0.842 | 0.693 | 0.667 |
| CLM | 0.772 | 0.668 | 0.722 | 0.598 | 0.340 | 0.310 | 0.349 | 0.178 | 0.299 | 0.211 | 0.092 | 0.160 | 0.834 | 0.592 | 0.710 | 0.861 | 0.612 | 0.803 | 0.089 | 0.186 | 0.207 | 0.818 | 0.693 | 0.642 |
| FUSE (ours) | 0.883 | 0.761 | 0.796 | 0.644 | 0.372 | 0.489 | 0.437 | 0.218 | 0.311 | 0.269 | 0.094 | 0.200 | 0.945 | 0.945 | 0.974 | 0.921 | 0.944 | 0.941 | 0.037 | 0.037 | 0.041 | 0.940 | 0.770 | 0.793 |
| Logistic $u^\star$ | 0.877 | 0.751 | 0.785 | 0.631 | 0.366 | 0.480 | 0.429 | 0.214 | 0.305 | 0.244 | 0.091 | 0.189 | 0.934 | 0.941 | 0.959 | 0.899 | 0.939 | 0.939 | 0.039 | 0.039 | 0.042 | 0.926 | 0.767 | 0.788 |
| Mean-only $u^\star$ | 0.880 | 0.757 | 0.792 | 0.642 | 0.369 | 0.479 | 0.433 | 0.212 | 0.307 | 0.252 | 0.092 | 0.192 | 0.941 | 0.943 | 0.963 | 0.919 | 0.940 | 0.940 | 0.039 | 0.038 | 0.044 | 0.931 | 0.769 | 0.769 |
| Quantile $\theta^\star$ | 0.881 | 0.757 | 0.788 | 0.628 | 0.370 | 0.480 | 0.428 | 0.213 | 0.300 | 0.251 | 0.090 | 0.188 | 0.929 | 0.941 | 0.958 | 0.899 | 0.935 | 0.929 | 0.042 | 0.040 | 0.044 | 0.930 | 0.766 | 0.779 |
| Global Tuning | 0.872 | 0.752 | 0.774 | 0.628 | 0.369 | 0.477 | 0.427 | 0.209 | 0.295 | 0.237 | 0.090 | 0.186 | 0.911 | 0.943 | 0.970 | 0.891 | 0.934 | 0.934 | 0.047 | 0.044 | 0.041 | 0.919 | 0.763 | 0.789 |
| OLS $\tilde{V}$ | 0.872 | 0.751 | 0.779 | 0.629 | 0.371 | 0.478 | 0.430 | 0.208 | 0.297 | 0.244 | 0.091 | 0.188 | 0.935 | 0.943 | 0.961 | 0.902 | 0.937 | 0.926 | 0.044 | 0.042 | 0.042 | 0.918 | 0.765 | 0.782 |
| Unweighted | 0.864 | 0.750 | 0.774 | 0.631 | 0.368 | 0.480 | 0.430 | 0.210 | 0.296 | 0.250 | 0.091 | 0.190 | 0.939 | 0.944 | 0.960 | 0.915 | 0.940 | 0.927 | 0.039 | 0.040 | 0.043 | 0.923 | 0.764 | 0.790 |

*Table 1.* Uncertainty evaluation on VQAv2, OKVQA, and AdVQA. We report AUROC, TPR at 0.1, 0.05, and 0.01 FPR, calibration correlations (Pearson, Spearman), Expected Calibration Error (ECE; lower is better), and AURAC (higher is better).

**Results.** As shown in Table 1, FUSE consistently achieves the highest AURAC across datasets. This indicates that fusing dual-source uncertainty yields a more reliable abstention signal: the model preferentially rejects queries that are likely to be incorrect, thereby improving accuracy on the answered subset. In practical VQA settings, this supports user-facing behaviors where the system abstains when uncertain rather than returning low-confidence answers. Qualitative results in Appendix L also suggest that FUSE is able to calibrate itself to output low scores for correct/certain output and vice versa.

### 5.4. Ablation Studies

We discuss some design choices of FUSE by ablating how we construct a single fused uncertainty statistic $u^\star$ from the posterior, and how we select the global threshold $\theta^\star$.

**Logistic** $u^\star$: we fit a logistic function on a calibration dataset, for each query $q$ we form $m_{\text{post},q}$ and $\sigma_{\text{post},q}$ and fit $u_q^\star = \Pr(y_q{=}1 \mid m_{\text{post},q}, \sigma_{\text{post},q}) = \sigma(\gamma_0 + \gamma_1 m_{\text{post},q} + \gamma_2 \log \sigma_{\text{post},q})$, using logistic regression and $y_q \in \{0,1\}$ indicates an incorrect outcome. **Mean-only** $u^\star$: we test using $u^\star = m_{\text{post}}$. **Quantile** $\theta^\star$: we choose the global threshold by enforcing a coverage constraint: $\theta^\star = \inf_\theta \left\{ \frac{1}{N} \sum_q \mathbf{1}\{u_q^\star \le \theta\} \ge \rho \right\}$, where $\theta^\star$ is the empirical $\rho$-quantile: $\theta^\star = \text{Quantile}_\rho\left(\{u_q^\star\}_{q=1}^N\right)$. **Global Tuning**: we tune $\alpha_0, \beta_0, \sigma_0^2, a, \tau^2$ involved in the Bayesian inference process together instead of analytically computing some of them. When labeled correctness indicators $y_i \in \{0,1\}$ are available, the parameters can be jointly tuned by minimizing an empirical risk over the posterior uncertainty scores $m_{\text{post},i}$: $\min_{\alpha_0,\beta_0,\sigma_0^2,a,\tau^2} \sum_i \ell\left(y_i, m_{\text{post},i}(\alpha_0, \beta_0, \sigma_0^2, a, \tau^2)\right)$, where $\ell$ is the negative log-likelihood. **OLS** $\tilde{V}$: we compute $\tilde{V}$ and fit the intercept $a$ and variance $\tau^2$ in $\tilde{V}_i = a + d\,\hat{u}_i + \varepsilon_i, \ \varepsilon_i \sim \mathcal{N}(0, \tau^2)$, using least squares: $\hat{a} = \bar{\tilde{V}} - d\,\bar{\hat{u}}, \ \hat{\tau}^2 = \frac{1}{N-2} \sum_i (\tilde{V}_i - \hat{a} - d\,\hat{u}_i)^2$. **Unweighted**: we measure the performance of $u^\star$ obtained from Theorem 4.4, where the semantic evidence is unweighted.

**Results.** As shown in Table 1, most variants of FUSE re-sulted in better performance than the baselines, illustrating the importance of the fused sources of uncertainty. Interestingly, with incoherence weights and a closed-form posterior with unknown parameters, Global Tuning remains a strong baseline, even though the parameters are learned. This implies the significance of using the posterior (*i.e.*, both sources) for the final uncertainty. More ablations on #generation, uncertainty sources, and evaluation criterion, and ablation details are shown in Appendix J.

### 5.5. Implementation Details

We use CLIP-ViT-L-336px as the backbone for prior and the embedding dimension $D = 768$. To train the GP adapters for the CLIP heads, we use a dimension of $Q = 10$ and 250 inducing points. We train the adapters on a combination of MS-COCO (Lin et al., 2014), Flickr30k (Plummer et al., 2015), and VQAv2 (Goyal et al., 2017) training datasets using AdamW, learning rate of 1e-6, and a batch size of 128 for 200 epochs. We hold out a calibration set of 500 samples for each experiment. For each baseline and query, we sample $n = 50$ stochastic responses using MCT (Cecere et al., 2025). We follow (Lau et al., 2025) and use `Exact-Match` to evaluate the correctness of VQA results. For the risk tolerance $r$ in calibration, we use $r = 0.2$.

## 6. Conclusion

We presented FUSE, an approach for rigorously quantifying uncertainty in VLMs. Our method probes into two complementary sources of uncertainty in VLMs: *aleatoric* uncertainty from input data embeddings and *epistemic* uncertainty from the model itself. Using a separate adapter for the input embedding without retraining the VLM, we probabilistically model the two sources of uncertainty and use a principled Bayesian inference procedure to combine them, yielding a single closed-form uncertainty statistic, approximating the uncertainty associated with VLM output for each query, for downstream calibration. Empirical results suggest that FUSE outperforms other baselines on uncertainty calibration and selective answering on multiple datasets. We discuss the limitations of FUSE in Appendix M.

## Acknowledgements

This work was partially funded by ONR RAPID Program and by Carlone's NSF CAREER Award.

## Impact Statement

This paper presents work whose goal is to advance the field of machine learning. There are many potential societal consequences of our work, none of which we feel must be specifically highlighted here.

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

# Supplementary Material

## A. Justification for the Aleatoric-Epistemic Decomposition

We define the variables that appear explicitly in our pipeline:

- $\boldsymbol{Z}$: observed encoder embedding produced by a frozen Vision-Language encoder (*e.g.*, CLIP).
- $\boldsymbol{x}$: latent representation inferred by the GP adapter from $\boldsymbol{Z}$, estimating the true representation of $\boldsymbol{Z}$.
- $\boldsymbol{r}$: random response embedding produced by the downstream VLM.

GP adapter defines a posterior $p(\boldsymbol{x} \mid \boldsymbol{Z})$ capturing encoder-level uncertainty, and the MLLM induces a conditional distribution $p_\theta(\boldsymbol{r} \mid \boldsymbol{x})$ over responses.

We assume that the latent $\boldsymbol{x}$ is drawn from

$$\boldsymbol{x} \sim p(\boldsymbol{x} \mid \boldsymbol{Z}),$$

which aggregates all encoder-induced uncertainty given $\boldsymbol{Z}$. Conditioned on $\boldsymbol{x}$, the MLLM generates responses via

$$\boldsymbol{r} \mid \boldsymbol{x}, \theta \sim p_\theta(\boldsymbol{r} \mid \boldsymbol{x}),$$

with fixed parameters $\theta$. In particular, $\boldsymbol{r}$ depends on $\boldsymbol{Z}$ only through $\boldsymbol{x}$, i.e. we have the Markov chain:

$$\boldsymbol{Z} \;\to\; \boldsymbol{x} \;\to\; \boldsymbol{r}. \tag{9}$$

Under the above assumptions, we have:

**Theorem A.1** (Encoder-level aleatoric-epistemic decomposition)**.** *The conditional variance of the response embedding given the encoder output satisfies*

$$\mathrm{Var}[\boldsymbol{r} \mid \boldsymbol{Z}] \;=\; \underbrace{\mathbb{E}_{\boldsymbol{x}\mid\boldsymbol{Z}}\big[\mathrm{Var}[\boldsymbol{r} \mid \boldsymbol{x}]\big]}_{\text{epistemic uncertainty}} \;+\; \underbrace{\mathrm{Var}_{\boldsymbol{x}\mid\boldsymbol{Z}}\big[\mathbb{E}[\boldsymbol{r} \mid \boldsymbol{x}]\big]}_{\text{aleatoric uncertainty}}. \tag{10}$$

*Proof.* By Equation (9), $\boldsymbol{r}$ depends on $\boldsymbol{Z}$ only through $\boldsymbol{x}$, so the law of total variance with respect to $\boldsymbol{x} \mid \boldsymbol{Z}$ gives

$$\mathrm{Var}[\boldsymbol{r} \mid \boldsymbol{Z}] = \mathbb{E}_{\boldsymbol{x}\mid\boldsymbol{Z}}\big[\mathrm{Var}[\boldsymbol{r} \mid \boldsymbol{x}, \boldsymbol{Z}]\big] \;+\; \mathrm{Var}_{\boldsymbol{x}\mid\boldsymbol{Z}}\big[\mathbb{E}[\boldsymbol{r} \mid \boldsymbol{x}, \boldsymbol{Z}]\big].$$

Since $\boldsymbol{r} \mid \boldsymbol{x}, \boldsymbol{Z}$ and $\boldsymbol{r} \mid \boldsymbol{x}$ have the same distribution (Markov chain $\boldsymbol{Z} \to \boldsymbol{x} \to \boldsymbol{r}$), we obtain

$$\mathrm{Var}[\boldsymbol{r} \mid \boldsymbol{Z}] = \mathbb{E}_{\boldsymbol{x}\mid\boldsymbol{Z}}\big[\mathrm{Var}[\boldsymbol{r} \mid \boldsymbol{x}]\big] \;+\; \mathrm{Var}_{\boldsymbol{x}\mid\boldsymbol{Z}}\big[\mathbb{E}[\boldsymbol{r} \mid \boldsymbol{x}]\big],$$

which is exactly (10). $\qquad\square$

The second term in (10) depends only on the posterior $p(\boldsymbol{x} \mid \boldsymbol{Z})$ defined by the GP adapter. It therefore quantifies *aleatoric uncertainty* arising from the encoder and the inherent ambiguity of the multimodal input. The first term measures residual variation of $\boldsymbol{r}$ when $\boldsymbol{x}$ is fixed; this is precisely the *epistemic uncertainty* via the semantic evidence of MLLM responses conditioned on $\boldsymbol{x}$.

## B. Proof of Theorem 4.4

**Theorem 4.4** (Linearity of Semantic Evidence in $u$)**.** *Given the unweighted semantic scatter* $\mathbf{S}$ *and its corresponding semantic evidence* $\tilde{V}$ *in Definition 4.3, we then have:*

$$\tilde{V} \;=\; n\,u \;+\; \log\det\boldsymbol{\Lambda}, \tag{5}$$

*where* $\boldsymbol{\Lambda} \sim \mathsf{Wishart}_n(\boldsymbol{I}, d)$ *is a $n$-dimensional Wishart random matrix.*

*Proof.* Write the unique symmetric square root $\boldsymbol{\Sigma}_r^{1/2}$ so that $\boldsymbol{\Sigma}_r = \boldsymbol{\Sigma}_r^{1/2}\boldsymbol{\Sigma}_r^{1/2}$. For each $i$, define

$$\boldsymbol{\omega}_i \;:=\; e^{-u/2}\,\boldsymbol{\Sigma}_r^{-1/2}\,\mathbf{r}_i.$$

Since $\mathbf{r}_i \sim \mathcal{N}(0, e^u \boldsymbol{\Sigma}_r)$, it follows that $\boldsymbol{\omega}_i \sim \mathcal{N}(0, \boldsymbol{I}_d)$ and $\{\boldsymbol{\omega}_i\}_{i=1}^n$ are i.i.d.

Stack the row vectors into the matrices

$$
\boldsymbol{R} := \begin{bmatrix} \boldsymbol{r}_1^\top \\ \vdots \\ \boldsymbol{r}_n^\top \end{bmatrix} \in \mathbb{R}^{n \times d}, \qquad \boldsymbol{\Omega} := \begin{bmatrix} \boldsymbol{\omega}_1^\top \\ \vdots \\ \boldsymbol{\omega}_n^\top \end{bmatrix} \in \mathbb{R}^{n \times d}.
$$

By construction,

$$
\boldsymbol{R} = e^{u/2}\, \boldsymbol{\Omega}\, \boldsymbol{\Sigma}_r^{1/2}.
$$

We define the (unweighted) *semantic Gram matrix* as

$$
\mathbf{S} := \boldsymbol{R}\boldsymbol{R}^\top \in \mathbb{R}^{n \times n}.
$$

Substituting $\boldsymbol{R} = e^{u/2}\boldsymbol{\Omega}\boldsymbol{\Sigma}_r^{1/2}$ yields

$$
\mathbf{S} = \left(e^{u/2}\boldsymbol{\Omega}\boldsymbol{\Sigma}_r^{1/2}\right)\left(e^{u/2}\boldsymbol{\Omega}\boldsymbol{\Sigma}_r^{1/2}\right)^\top = e^u\, \boldsymbol{\Omega}\,\boldsymbol{\Sigma}_r\,\boldsymbol{\Omega}^\top.
$$

Taking determinants and using $\det(e^u \boldsymbol{A}) = e^{un}\det(\boldsymbol{A})$ for $\boldsymbol{A} \in \mathbb{R}^{n \times n}$ gives

$$
\det \mathbf{S} = e^{un}\, \det\!\left(\boldsymbol{\Omega}\,\boldsymbol{\Sigma}_r\,\boldsymbol{\Omega}^\top\right).
$$

Applying $\log$ yields

$$
\log \det \mathbf{S} = n\,u + \log\det\!\left(\boldsymbol{\Omega}\,\boldsymbol{\Sigma}_r\,\boldsymbol{\Omega}^\top\right).
$$

In the whitened setting $\boldsymbol{\Sigma}_r = \boldsymbol{I}_d$ (see Appendix G), the above reduces to

$$
\log \det \mathbf{S} = n\,u + \log\det(\boldsymbol{\Omega}\boldsymbol{\Omega}^\top).
$$

Finally, since $\boldsymbol{\Omega}$ has i.i.d. $\mathcal{N}(0,1)$ entries, we have $\boldsymbol{\Omega}\boldsymbol{\Omega}^\top \sim \mathsf{Wishart}_n(\boldsymbol{I}, d)$, which we denote by $\boldsymbol{\Lambda}$. Therefore,

$$
\tilde{V} = \log \det \mathbf{S} = n\,u + \log\det \boldsymbol{\Lambda},
$$

establishing exact linearity in $u$ with slope $n$.

$\square$

## C. Proof of Theorem 4.5

**Lemma C.1** (Moments of $\log \chi_k^2$). *Let $X \sim \chi_k^2$. Then*

$$
\mathbb{E}[\log X] = \psi\!\left(\tfrac{k}{2}\right) + \log 2, \qquad \mathrm{Var}[\log X] = \psi_1\!\left(\tfrac{k}{2}\right), \tag{11}
$$

*where $\psi$ and $\psi_1$ denote the digamma and trigamma functions, respectively.*

*Proof.* Recall that if $X \sim \chi_k^2$, then by definition $X$ can be expressed as a scaled Gamma random variable:

$$
X = 2Y, \qquad Y \sim \mathrm{Gamma}\!\left(\frac{k}{2}, 1\right), \tag{12}
$$

where the Gamma distribution is parameterized by shape $\alpha = k/2$ and scale $\theta = 1$, with density

$$
f_Y(y) = \frac{1}{\Gamma(\alpha)} y^{\alpha-1} e^{-y}, \qquad y > 0. \tag{13}
$$

Using this transformation, we have

$$
\log X = \log(2Y) = \log 2 + \log Y. \tag{14}
$$

Since $\log 2$ is constant, we can write

$$\mathbb{E}[\log X] = \log 2 + \mathbb{E}[\log Y], \qquad \mathrm{Var}[\log X] = \mathrm{Var}[\log Y]. \tag{15}$$

To compute $\mathbb{E}[\log Y]$ and $\mathrm{Var}[\log Y]$, we exploit the moment generating relation of the Gamma distribution. For any real $t$ such that $\alpha + t > 0$, the $t$-th moment of $Y$ is

$$\mathbb{E}[Y^t] = \frac{\Gamma(\alpha + t)}{\Gamma(\alpha)}. \tag{16}$$

Taking the natural logarithm of both sides gives

$$\log \mathbb{E}[Y^t] = \log \Gamma(\alpha + t) - \log \Gamma(\alpha). \tag{17}$$

Differentiating both sides with respect to $t$ and evaluating at $t = 0$ yields

$$\frac{d}{dt} \log \mathbb{E}[Y^t]\Big|_{t=0} = \frac{d}{dt} \log \Gamma(\alpha + t)\Big|_{t=0} = \psi(\alpha), \tag{18}$$

where $\psi(\cdot)$ is the *digamma function*, defined as the logarithmic derivative of the Gamma function:

$$\psi(\alpha) = \frac{d}{d\alpha} \log \Gamma(\alpha) = \frac{\Gamma'(\alpha)}{\Gamma(\alpha)}. \tag{19}$$

The left-hand side corresponds to the derivative of $\mathbb{E}[Y^t]$ at $t = 0$:

$$\frac{d}{dt} \mathbb{E}[Y^t]\Big|_{t=0} = \mathbb{E}[\log Y], \tag{20}$$

Hence we obtain

$$\mathbb{E}[\log Y] = \psi(\alpha). \tag{21}$$

Similarly, differentiating twice gives

$$\frac{d^2}{dt^2} \log \mathbb{E}[Y^t]\Big|_{t=0} = \psi_1(\alpha), \tag{22}$$

where $\psi_1(\cdot)$ denotes the *trigamma function*, i.e. the first derivative of the digamma function:

$$\psi_1(\alpha) = \frac{d}{d\alpha} \psi(\alpha). \tag{23}$$

This quantity is the variance of $\log Y$, since

$$\mathrm{Var}[\log Y] = \mathbb{E}[(\log Y)^2] - \mathbb{E}[\log Y]^2 = \psi_1(\alpha). \tag{24}$$

Finally, substituting $\alpha = k/2$ and recalling that $\log X = \log 2 + \log Y$, we obtain

$$\mathbb{E}[\log X] = \psi\left(\frac{k}{2}\right) + \log 2, \qquad \mathrm{Var}[\log X] = \psi_1\left(\frac{k}{2}\right), \tag{25}$$

which completes the proof. $\square$

**Theorem 4.5** (Gaussian Likelihood of Semantic Evidence). *The semantic evidence $\tilde{V}$ defined in Definition 4.3, for given $d$ and $n$, follows $\tilde{V} \mid u \sim \mathcal{N}\left(a(n, d) + n\, u,\ v(n, d)\right)$, where $a(n, d)$ and $v(n, d)$ are analytical functions of $n$ and $d$ defined in Appendix C.*

*Proof.* Recall from Theorem 4.4 that in the unweighted case we define the semantic Gram matrix

$$\mathbf{S} := \boldsymbol{R}\boldsymbol{R}^\top \in \mathbb{R}^{n \times n}, \qquad \tilde{V} := \log \det \mathbf{S}. \tag{26}$$

With response embedding whitening, we may write

$$\boldsymbol{R} = e^{u/2}\,\boldsymbol{\Omega}, \qquad \boldsymbol{\Omega} \in \mathbb{R}^{n \times d} \text{ has i.i.d. } \mathcal{N}(0,1) \text{ entries,} \tag{27}$$

and hence

$$\mathbf{S} = \boldsymbol{R}\boldsymbol{R}^\top = e^u\,\boldsymbol{\Omega}\boldsymbol{\Omega}^\top = e^u\,\boldsymbol{\Lambda}, \tag{28}$$

where $\boldsymbol{\Lambda} \sim \mathrm{Wishart}_n(\boldsymbol{I}, d)$ is an $n$-dimensional Wishart random matrix with $d$ degrees of freedom. Therefore,

$$\tilde{V} = \log \det \mathbf{S} = \log \det(e^u \boldsymbol{\Lambda}) = n\,u + \log \det \boldsymbol{\Lambda}. \tag{29}$$

Recall the Bartlett decomposition of the Wishart distribution, there exists an upper-triangular matrix $\boldsymbol{T} \in \mathbb{R}^{n \times n}$ such that $\boldsymbol{\Lambda} = \boldsymbol{T}^\top \boldsymbol{T}$ and the entries satisfy

$$\begin{cases} \boldsymbol{T}_{jj}^2 \sim \chi_{d-j+1}^2, & j = 1, \ldots, n, \\ \boldsymbol{T}_{ij} \sim \mathcal{N}(0,1), & 1 \le i < j \le n, \end{cases} \tag{30}$$

with all $\{\boldsymbol{T}_{jj}^2\}_{j=1}^n$ mutually independent and independent of the off-diagonal Gaussians. Consequently,

$$\det \boldsymbol{\Lambda} = \prod_{j=1}^n \boldsymbol{T}_{jj}^2, \qquad \log \det \boldsymbol{\Lambda} = \sum_{j=1}^n \log \boldsymbol{T}_{jj}^2. \tag{31}$$

Equivalently, there exist independent $\chi^2$ random variables $\{\xi_j\}_{j=1}^n$ with

$$\xi_j \sim \chi_{d+1-j}^2 \qquad (j = 1, \ldots, n), \tag{32}$$

such that $\log \det \boldsymbol{\Lambda} \overset{d}{=} \sum_{j=1}^n \log \xi_j$.

Via Lemma C.1, taking expectations and variances and summing over $j$ gives

$$\mathbb{E}[\log \det \boldsymbol{\Lambda}] = \sum_{j=1}^n \left\{ \psi\!\left( \tfrac{d+1-j}{2} \right) + \log 2 \right\}, \tag{33}$$

$$\mathrm{Var}[\log \det \boldsymbol{\Lambda}] = \sum_{j=1}^n \psi_1\!\left( \tfrac{d+1-j}{2} \right). \tag{34}$$

Thus, using $\tilde{V} = n\,u + \log \det \boldsymbol{\Lambda}$ and linearity of expectation,

$$\mathbb{E}[\tilde{V} \mid u] = n\,u + \mathbb{E}[\log \det \boldsymbol{\Lambda}] = a(d, n) + n\,u, \tag{35}$$

and

$$\mathrm{Var}[\tilde{V} \mid u] = \mathrm{Var}[\log \det \boldsymbol{\Lambda}] = v(d, n), \tag{36}$$

where we define

$$a(d, n) := \sum_{j=1}^n \left\{ \psi\!\left( \tfrac{d+1-j}{2} \right) + \log 2 \right\}, \qquad v(d, n) := \sum_{j=1}^n \psi_1\!\left( \tfrac{d+1-j}{2} \right). \tag{37}$$

Define centered summands

$$Y_j := \log \xi_j - \mathbb{E}[\log \xi_j], \qquad j = 1, \ldots, n, \tag{38}$$

which are independent with finite variances $\psi_1\big((d+1-j)/2\big)$ and finite third moments. Then

$$\log \det \boldsymbol{\Lambda} - \mathbb{E}[\log \det \boldsymbol{\Lambda}] = \sum_{j=1}^n Y_j. \tag{39}$$

By the Lindeberg–Feller CLT for independent, non-identically distributed random variables, as $d \to \infty$ with fixed $n$,

$$\frac{\sum_{j=1}^{n} Y_j}{\sqrt{\sum_{j=1}^{n} \mathrm{Var}(Y_j)}} = \frac{\log \det \mathbf{\Lambda} - \mathbb{E}[\log \det \mathbf{\Lambda}]}{\sqrt{v(d, n)}} \xrightarrow{d} \mathcal{N}(0, 1). \tag{40}$$

Combining with $\tilde{V} = n\,u + \log \det \mathbf{\Lambda}$ yields the stated Gaussian approximation for $\tilde{V} \mid u$ with mean $a(d, n) + n\,u$ and variance $v(d, n)$. $\qquad \square$

## D. Proof of Theorem 4.8

**Theorem 4.8** (Weighted Semantic Evidence Likelihood). *Assume the weights $\{w_i\}$ are bounded away from $0$ and $\infty$, Then for fixed $d$ and $n$, we have*

$$\tilde{V} = n\,u + a_w(\nu_{\mathrm{eff}}, d, n, \bar{w}) + \varepsilon_w, \tag{6}$$

*where $\varepsilon_w \sim \mathcal{N}\left(0, \ v(d, n)\right)$. Here, $\bar{w} = \frac{1}{n} \sum_{i=1}^{n} w_i$, $\nu_{\mathrm{eff}} = \frac{(\sum_i w_i)^2}{\sum_i w_i^2}$, where $a_w(n, d)$ and $v(n, d)$ are analytical functions of $n, d$ defined in Appendix D.*

*Proof.* Since $\boldsymbol{r}_i \overset{\text{i.i.d.}}{\sim} \mathcal{N}(0, e^u \boldsymbol{I}_d)$, we may write

$$\boldsymbol{R} = e^{u/2} \boldsymbol{\Omega}, \qquad \boldsymbol{\Omega} \in \mathbb{R}^{n \times d} \text{ has i.i.d. } \mathcal{N}(0, 1) \text{ entries.}$$

Hence the weighted scatter matrix satisfies

$$\boldsymbol{S}_w = \boldsymbol{W}^{1/2} \boldsymbol{R} \boldsymbol{R}^{\top} \boldsymbol{W}^{1/2} = e^u\, \boldsymbol{W}^{1/2} \boldsymbol{\Omega} \boldsymbol{\Omega}^{\top} \boldsymbol{W}^{1/2}.$$

Taking determinants (noting $\det(e^u \boldsymbol{A}) = e^{un} \det(\boldsymbol{A})$ for $\boldsymbol{A} \in \mathbb{R}^{n \times n}$) gives

$$\tilde{V} = \log \det(\boldsymbol{S}_w) = n\,u + \log \det\left( \boldsymbol{W}^{1/2} \boldsymbol{\Omega} \boldsymbol{\Omega}^{\top} \boldsymbol{W}^{1/2} \right). \tag{41}$$

Using $\det(\boldsymbol{W}^{1/2} \boldsymbol{A} \boldsymbol{W}^{1/2}) = \det(\boldsymbol{W}) \det(\boldsymbol{A})$ for square $\boldsymbol{A} \in \mathbb{R}^{n \times n}$, we obtain the exact decomposition

$$\tilde{V} = n\,u + \log \det(\boldsymbol{W}) + \log \det(\boldsymbol{\Omega} \boldsymbol{\Omega}^{\top}). \tag{42}$$

**Wishart term.** Since $\boldsymbol{\Omega}$ is standard Gaussian, $\boldsymbol{\Omega} \boldsymbol{\Omega}^{\top} \sim \mathrm{Wishart}_n(\boldsymbol{I}, d)$. By Bartlett decomposition, there exist independent $\chi^2$ random variables $\{\xi_j\}_{j=1}^{n}$ with

$$\xi_j \sim \chi_{d+1-j}^2 \qquad (j = 1, \dots, n)$$

such that $\log \det(\boldsymbol{\Omega} \boldsymbol{\Omega}^{\top}) \overset{d}{=} \sum_{j=1}^{n} \log \xi_j$. Therefore, by Lemma C.1,

$$\mathbb{E}\left[ \log \det(\boldsymbol{\Omega} \boldsymbol{\Omega}^{\top}) \right] = \sum_{j=1}^{n} \left\{ \psi\left( \tfrac{d+1-j}{2} \right) + \log 2 \right\} = a_0(d, n), \tag{43}$$

$$\mathrm{Var}\left[ \log \det(\boldsymbol{\Omega} \boldsymbol{\Omega}^{\top}) \right] = \sum_{j=1}^{n} \psi_1\left( \tfrac{d+1-j}{2} \right) = v(d, n). \tag{44}$$

Moreover, for fixed $n$, a normal approximation for $\sum_{j=1}^{n} \log \xi_j$ follows as $d \to \infty$ (e.g., via a smooth-function expansion of $\log \chi_k^2$ around its mean and the Lindeberg–Feller CLT in the growing-d.f. regime), yielding

$$\log \det(\boldsymbol{\Omega} \boldsymbol{\Omega}^{\top}) \overset{\mathrm{approx}}{\sim} \mathcal{N}\left( a_0(d, n), \ v(d, n) \right).$$

**Weight term (expressed via $\bar{w}$ and $\nu_{\text{eff}}$).** Write $w_i = \bar{w}(1 + \delta_i)$ where $\delta_i := (w_i - \bar{w})/\bar{w}$ satisfies $\sum_{i=1}^{n} \delta_i = 0$. A second-order Taylor expansion of $\log(1 + \delta)$ gives

$$\sum_{i=1}^{n} \log w_i = n \log \bar{w} + \sum_{i=1}^{n} \log(1 + \delta_i) = n \log \bar{w} - \frac{1}{2} \sum_{i=1}^{n} \delta_i^2 + R_w,$$

where the remainder satisfies $|R_w| \leq C \sum_{i=1}^{n} |\delta_i|^3$ for a constant $C$ depending only on the bound $w_{\min} \leq w_i \leq w_{\max}$ (hence $|\delta_i|$ is uniformly bounded). Next observe that

$$\kappa_w = \frac{\sum_i w_i^2}{(\sum_i w_i)^2} = \frac{\sum_i \bar{w}^2 (1 + \delta_i)^2}{(n\bar{w})^2} = \frac{1}{n}\left(1 + \frac{1}{n} \sum_{i=1}^{n} \delta_i^2\right),$$

so that

$$\sum_{i=1}^{n} \delta_i^2 = n\left(n\kappa_w - 1\right) = n\left(\frac{n}{\nu_{\text{eff}}} - 1\right).$$

Substituting back yields

$$\log \det(\boldsymbol{W}) = \sum_{i=1}^{n} \log w_i = n \log \bar{w} - \frac{n}{2}\left(\frac{n}{\nu_{\text{eff}}} - 1\right) + R_w. \tag{45}$$

(The remainder $R_w$ is a higher-order dispersion term; under bounded weights it is controlled by the same dispersion statistics and is typically small when weights are not extremely heterogeneous.)

**Collecting terms.** Plugging (45) and the Wishart approximation into the exact decomposition (42) gives

$$\tilde{V} = n\,u + \left[a_0(d, n) + n \log \bar{w} - \frac{n}{2}\left(\frac{n}{\nu_{\text{eff}}} - 1\right)\right] + \underbrace{\left(\log \det(\boldsymbol{\Omega\Omega}^\top) - a_0(d, n)\right)}_{\varepsilon_w} + R_w.$$

Absorbing the (typically small) remainder $R_w$ into the intercept approximation yields the stated form $\tilde{V} = nu + a_w(\nu_{\text{eff}}, d, n, \bar{w}) + \varepsilon_w$ with $\varepsilon_w \overset{\text{approx}}{\sim} \mathcal{N}(0, v(d, n))$, where

$$a_w(\nu_{\text{eff}}, d, n, \bar{w}) := a_0(d, n) + n \log \bar{w} - \frac{n}{2}\left(\frac{n}{\nu_{\text{eff}}} - 1\right), \qquad v(d, n) := \sum_{j=1}^{n} \psi_1\left(\frac{d + 1 - j}{2}\right), \tag{46}$$

and

$$a_0(d, n) := \sum_{j=1}^{n} \psi\left(\frac{d + 1 - j}{2}\right) + n \log 2, \tag{47}$$

$\square$

# E. Proof of Theorem 4.9

**Theorem 4.9** (Closed-Form Posterior of Uncertainty). *Combining the data-representation prior (Definition 4.1) with the linear-Gaussian evidence model gives*

$$u \mid \tilde{V} \sim \mathcal{N}\left(m_{\text{post}}, \sigma_{\text{post}}^2\right), \tag{8}$$

*where* $\sigma_{\text{post}}^2 = \left(\frac{1}{\sigma_0^2} + \frac{n^2}{\tau^2}\right)^{-1}, m_{\text{post}} = \sigma_{\text{post}}^2\left(\frac{m_0}{\sigma_0^2} + \frac{n(\tilde{V}-a)}{\tau^2}\right)$, *with* $a, \tau^2$ *given in Equation* (7).

*Proof.* By Bayes' rule,

$$p(u \mid \tilde{V}) \propto p(\tilde{V} \mid u)\,p(u). \tag{48}$$

With the stated Gaussians,

$$p(\tilde{V} \mid u) = \frac{1}{\sqrt{2\pi\tau^2}} \exp\left(-\frac{1}{2\tau^2}(\tilde{V} - a - nu)^2\right), \qquad p(u) = \frac{1}{\sqrt{2\pi\sigma_0^2}} \exp\left(-\frac{1}{2\sigma_0^2}(u - m_0)^2\right). \tag{49}$$

Taking logs and discarding constants independent of $u$,

$$\log p(u \mid \tilde{V}) = -\frac{1}{2\tau^2}(\tilde{V} - a - nu)^2 - \frac{1}{2\sigma_0^2}(u - m_0)^2 + \text{const} \tag{50}$$

$$= -\frac{1}{2}\Big(\frac{d^2}{\tau^2} + \frac{1}{\sigma_0^2}\Big)u^2 + \Big(\frac{d(\tilde{V}-a)}{\tau^2} + \frac{m_0}{\sigma_0^2}\Big)u + \text{const}. \tag{51}$$

Introduce $\lambda := \frac{n^2}{\tau^2} + \frac{1}{\sigma_0^2}$, $\eta := \frac{n(\tilde{V}-a)}{\tau^2} + \frac{m_0}{\sigma_0^2}$. Then

$$\log p(u \mid \tilde{V}) = -\frac{\lambda}{2}u^2 + \eta\, u + \text{const} = -\frac{\lambda}{2}\Big(u - \frac{\eta}{\lambda}\Big)^2 + \frac{\eta^2}{2\lambda} + \text{const}, \tag{52}$$

where we completed the square using

$$u^2 - 2(\eta/\lambda)u = (u - \eta/\lambda)^2 - (\eta/\lambda)^2. \tag{53}$$

Exponentiating and absorbing constants into a factor $Z^{-1}$,

$$p(u \mid \tilde{V}) = Z^{-1}\exp\Big(-\frac{\lambda}{2}\Big(u - \frac{\eta}{\lambda}\Big)^2\Big). \tag{54}$$

The normalizing constant is the Gaussian integral $Z = \int_{\mathbb{R}} \exp\big(-\frac{\lambda}{2}(u - \eta/\lambda)^2\big)\, du$. Hence

$$p(u \mid \tilde{V}) \sim \mathcal{N}\Big(\frac{\eta}{\lambda},\ \lambda^{-1}\Big). \tag{55}$$

Substituting back,

$$\sigma_{\text{post}}^2 = \lambda^{-1} = \Big(\frac{1}{\sigma_0^2} + \frac{n^2}{\tau^2}\Big)^{-1}, \qquad m_{\text{post}} = \frac{\eta}{\lambda} = \sigma_{\text{post}}^2\Big(\frac{m_0}{\sigma_0^2} + \frac{n(\tilde{V}-a)}{\tau^2}\Big). \tag{56}$$

Thus $u \mid \tilde{V}$ is Gaussian with the stated parameters.

$\square$

*Remark* E.1 (Monotonicity). Since $\partial m_{\text{post}}/\partial\tilde{V} = \frac{n}{\tau^2}\sigma_{\text{post}}^2 > 0$, the posterior mean increases monotonically with $\tilde{V}$. Moreover,

$$\text{as } \sigma_0^2 \to \infty \text{ (uninformative prior)}, \quad m_{\text{post}} \to \hat{u}_E,\ \sigma_{\text{post}}^2 \to \tau^2/n^2; \tag{57}$$

and

$$\text{as } \tau^2 \to \infty \text{ (uninformative evidence)}, \quad m_{\text{post}} \to m_0,\ \sigma_{\text{post}}^2 \to \sigma_0^2. \tag{58}$$

## F. Prior Calibration

Here we show how we calibrate the input-driven prior parameters $m_0(s) = \alpha_0 + \beta_0 s$ *without* using any correctness- or success-based proxy.

For each calibration query $q \in \{1, \ldots, Q\}$, let $s_q \in \mathbb{R}$ denote an *input-derived* statistic computed from the learned GP. Let $\tilde{V}_q$ denote the *semantic evidence* computed from sampled outputs. We assume the linear-Gaussian hierarchical model

$$u_q \mid s_q \sim \mathcal{N}\big(\alpha_0 + \beta_0 s_q,\ \sigma_0^2\big), \tag{59}$$

$$\tilde{V}_q \mid u_q \sim \mathcal{N}\big(a_q + b_q u_q,\ \tau_q^2\big), \tag{60}$$

where $u_q$ is the latent uncertainty scale, $b_q$ is the Gram slope (typically $b_q = n_q$), and $a_q, \tau_q^2$ are likelihood constants (e.g., from the Wishart/Bartlett moments; $a_q$ may include known deterministic terms such as $\log\det W_q$ when weights are used).

Since (59)–(60) are conjugate Gaussians, we can integrate out $u_q$ in closed form:

$$\tilde{V}_q \mid s_q \sim \mathcal{N}\big(a_q + b_q(\alpha_0 + \beta_0 s_q),\ \tau_q^2 + b_q^2\sigma_0^2\big). \tag{61}$$

Therefore, the calibration parameters $(\alpha_0, \beta_0, \sigma_0^2)$ can be estimated by maximizing the marginal log-likelihood

$$\mathcal{L}(\alpha_0, \beta_0, \sigma_0^2) \;=\; \sum_{q=1}^{Q} \log \mathcal{N}\Big(\tilde{V}_q \,;\, a_q + b_q(\alpha_0 + \beta_0 s_q), \; \tau_q^2 + b_q^2 \sigma_0^2\Big). \tag{62}$$

Crucially, (62) depends only on the input-derived statistic $s_q$ and the unsupervised dispersion statistic $\tilde{V}_q$; it does not require any correctness labels or empirical success probabilities.

For fixed $\sigma_0^2$, maximizing (62) over $(\alpha_0, \beta_0)$ reduces to a weighted least squares problem. Define

$$y_q \;:=\; \tilde{V}_q - a_q, \qquad x_q \;:=\; s_q, \qquad \omega_q(\sigma_0^2) \;:=\; \frac{b_q^2}{\tau_q^2 + b_q^2 \sigma_0^2}. \tag{63}$$

Then the negative log-likelihood (up to additive constants independent of $\alpha_0, \beta_0$) is

$$\sum_{q=1}^{Q} \omega_q(\sigma_0^2) \left( \tfrac{y_q}{b_q} - \alpha_0 - \beta_0 x_q \right)^2. \tag{64}$$

Let

$$\bar{x}_\omega \;:=\; \frac{\sum_q \omega_q x_q}{\sum_q \omega_q}, \qquad \bar{z}_\omega \;:=\; \frac{\sum_q \omega_q z_q}{\sum_q \omega_q}, \qquad z_q \;:=\; \frac{y_q}{b_q}. \tag{65}$$

The weighted least squares solution is

$$\hat{\beta}_0(\sigma_0^2) = \frac{\sum_{q=1}^{Q} \omega_q(\sigma_0^2)\,(x_q - \bar{x}_\omega)\,(z_q - \bar{z}_\omega)}{\sum_{q=1}^{Q} \omega_q(\sigma_0^2)\,(x_q - \bar{x}_\omega)^2}, \tag{66}$$

$$\hat{\alpha}_0(\sigma_0^2) = \bar{z}_\omega - \hat{\beta}_0(\sigma_0^2)\,\bar{x}_\omega. \tag{67}$$

We estimate $\sigma_0^2$ by maximizing (62) after substituting $(\alpha_0, \beta_0) = (\hat{\alpha}_0(\sigma_0^2), \hat{\beta}_0(\sigma_0^2))$ from (66)–(67). This yields a one-dimensional optimization in $\sigma_0^2 \geq 0$, which can be solved via a standard line search.

The calibration objective (62) uses only: (i) the input-derived statistic $s_q$ from the GP, and (ii) the unsupervised dispersion statistic $\tilde{V}_q$ computed from sampled outputs. It does not use task correctness, empirical success probabilities, or any label-derived proxy. As a result, the calibrated prior mean $m_0(s) = \alpha_0 + \beta_0 s$ remains input-driven and does not incorporate model-performance information, consistent with our intended separation of uncertainty sources.

## G. More on Response Evidence

### G.1. Assumption Justification

Our assumption, at its core, is a Gaussian scale family model. $\Sigma_r$ encodes the base geometry (orientation, anisotropy, correlation structure) of VLM response embeddings. The scalar $e^u$ encodes the overall magnitude or "volume" of uncertainty in semantic space. Sampling multiple responses from the MLLM corresponds to sampling from a distribution whose spread increases with uncertainty level. The Gaussian scale model is the cleanest closed-form model that captures the semantic evidence's isotropic scaling as well as mathematical tractability.

The assumption that response embeddings $r_i$ for a fixed multimodal query are distributed as a single zero-mean Gaussian (or more generally, an elliptical distribution) is empirically reasonable for LLaVA-style models. During instruction tuning, LLaVA is trained to produce a single canonical answer per input, leading to a sharply concentrated conditional distribution $p(r_i|\text{input})$. Therefore, modeling $r_i$ as Gaussian draws around a single latent mean captures the local geometry of the response manifold, whereas assuming multiple distinct semantic modes is neither common nor realistic for LLaVA or similar MLLMs.

### G.2. Estimation of the Response Covariance

We discuss how to practically obtain the base covariance $\boldsymbol{\Sigma}_r$. Given a set $\{\boldsymbol{r}_i^{(q)}\}_{q=1,\ldots,Q;\ i=1,\ldots,n_q}$ of stochastic responses sampled from the multimodal model when propmpted with a query $q$, we estimate $\boldsymbol{\Sigma}_r$ as the empirical covariance across all responses:

$$\widehat{\boldsymbol{\Sigma}}_r = \frac{1}{N}\sum_{q=1}^{Q}\sum_{i=1}^{n_q}\left(\boldsymbol{r}_i^{(q)} - \bar{\boldsymbol{r}}\right)\left(\boldsymbol{r}_i^{(q)} - \bar{\boldsymbol{r}}\right)^{\top}, \qquad \bar{\boldsymbol{r}} = \frac{1}{N}\sum_{q,i}\boldsymbol{r}_i^{(q)}, \tag{68}$$

where $N = \sum_q n_q$ is the total number of sampled responses. We remove the unknown scale $e^{u^{(q)}}$ by unit-trace normalization:

$$\widehat{\boldsymbol{\Sigma}}_r \ \leftarrow \ \frac{\widehat{\boldsymbol{\Sigma}}_r}{\operatorname{tr}(\widehat{\boldsymbol{\Sigma}}_r)/d}.$$

We then whiten all response embeddings by

$$\tilde{\boldsymbol{r}}_i^{(q)} = \widehat{\boldsymbol{\Sigma}}_r^{-1/2}\left(\boldsymbol{r}_i^{(q)} - \bar{\boldsymbol{r}}\right). \tag{69}$$

= After whitening, all empirical semantic-volume computations use $\tilde{\boldsymbol{r}}_i^{(q)}$ in place of $\boldsymbol{r}_i^{(q)}$.

## H. Decision Rule for Fused Uncertainty Statistic

We assume that alibration queries $\{(u_q^\star, e_q)\}_{q=1}^N$ are i.i.d. draws from the same distribution as test queries, where $e_q = \mathbf{1}\{\hat{y}_q \neq y_q\}$. Let $(U, E) \in \mathbb{R} \times \{0, 1\}$ denote a generic test query, where $U = u^\star$ is the fused uncertainty score and $E = \mathbf{1}\{\hat{y} \neq y\}$ is the error indicator. For any threshold $\theta \in \mathbb{R}$, define the *answered event* $A_\theta := \{U \leq \theta\}$, the *coverage* $C(\theta) := \mathbb{P}(A_\theta)$, and the *selective risk*

$$R(\theta) \ := \ \mathbb{P}(E = 1 \mid A_\theta) \ = \ \frac{\mathbb{P}(E = 1, A_\theta)}{\mathbb{P}(A_\theta)} \qquad (\text{whenever } \mathbb{P}(A_\theta) > 0). \tag{70}$$

Let $\{(U_q, E_q)\}_{q=1}^N$ be the held-out calibration set, assumed i.i.d.. For any $\theta$, define the answered count and the empirical selective risk:

$$m(\theta) := \sum_{q=1}^{N}\mathbf{1}\{U_q \leq \theta\}, \tag{71}$$

$$\hat{R}_N(\theta) := \frac{\sum_{q=1}^{N}\mathbf{1}\{U_q \leq \theta\}E_q}{m(\theta)} \qquad (\text{defined when } m(\theta) \geq 1). \tag{72}$$

Finally, let $\Theta_N$ be the finite set of candidate thresholds formed by the $N$ observed values of $\{U_q\}$, so that $|\Theta_N| \leq N + 1$. Fix $\delta \in (0, 1)$ and a minimum answer count $m_0 \geq 1$.

**Theorem H.1** (Finite-sample risk control up to estimation error)**.** *With probability at least* $1 - \delta$, *simultaneously for all* $\theta \in \Theta_N$ *such that* $\sum_{q=1}^{N}\mathbf{1}\{u_q^\star \leq \theta\} \geq m_0$,

$$R(\theta) \ \leq \ \hat{R}_N(\theta) \ + \ \sqrt{\frac{\log(2|\Theta_N|/\delta)}{2m_0}}.$$

*Consequently, if* $\theta^\star$ *is chosen by empirical risk control*

$$\theta^\star \in \arg\max_{\theta \in \Theta_N}\left\{\hat{C}_N(\theta) : \hat{R}_N(\theta) \leq r, \ \sum_{q=1}^{N}\mathbf{1}\{u_q^\star \leq \theta\} \geq m_0\right\},$$

*then with probability at least* $1 - \delta$,

$$R(\theta^\star) \ \leq \ r \ + \ \sqrt{\frac{\log(2|\Theta_N|/\delta)}{2m_0}}.$$

*Proof.* We prove that with probability at least $1 - \delta$, simultaneously for all $\theta \in \Theta_N$ with $m(\theta) \geq m_0$,

$$R(\theta) \leq \hat{R}_N(\theta) + \sqrt{\frac{\log(2|\Theta_N|/\delta)}{2m_0}}. \tag{73}$$

This implies the bound for $\theta^\star$ chosen by empirical risk control.

Fix any $\theta \in \mathbb{R}$ and consider the i.i.d. sequence

$$X_q(\theta) := \mathbf{1}\{U_q \leq \theta\}E_q \in \{0,1\}, \qquad Y_q(\theta) := \mathbf{1}\{U_q \leq \theta\} \in \{0,1\}.$$

Then $\sum_q Y_q(\theta) = m(\theta)$ and $\sum_q X_q(\theta)$ is the number of errors among answered queries.

Let $I(\theta) := \{q : U_q \leq \theta\}$ denote the (random) index set of answered points and note that

$$\hat{R}_N(\theta) = \frac{1}{m(\theta)} \sum_{q \in I(\theta)} E_q.$$

Condition on the event $\{m(\theta) = m\}$ for some integer $m \geq 1$. Under i.i.d. assumption, the answered examples are i.i.d. draws from the conditional distribution given $A_\theta = \{U \leq \theta\}$, hence the conditional distribution of $\{E_q\}_{q \in I(\theta)}$ is that of $m$ i.i.d. Bernoulli variables with mean $R(\theta) = \mathbb{P}(E = 1 \mid A_\theta)$. Therefore, conditional on $m(\theta) = m$,

$$\mathbb{P}\Big(R(\theta) - \hat{R}_N(\theta) \geq \varepsilon \,\Big|\, m(\theta) = m\Big) \leq \exp(-2m\varepsilon^2), \tag{74}$$

by Hoeffding's inequality.

Now restrict to thresholds with $m(\theta) \geq m_0$. On the event $\{m(\theta) \geq m_0\}$, we have $\exp(-2m\varepsilon^2) \leq \exp(-2m_0\varepsilon^2)$, hence taking expectation over $m(\theta)$ yields the unconditional bound

$$\mathbb{P}\Big(R(\theta) - \hat{R}_N(\theta) \geq \varepsilon \text{ and } m(\theta) \geq m_0\Big) \leq \exp(-2m_0\varepsilon^2). \tag{75}$$

We now apply a union bound over the finite set $\Theta_N$. For any $\varepsilon > 0$,

$$\mathbb{P}\Big(\exists \theta \in \Theta_N : R(\theta) - \hat{R}_N(\theta) \geq \varepsilon, \, m(\theta) \geq m_0\Big) \leq \sum_{\theta \in \Theta_N} \mathbb{P}\Big(R(\theta) - \hat{R}_N(\theta) \geq \varepsilon, \, m(\theta) \geq m_0\Big)$$
$$\leq |\Theta_N| \exp(-2m_0\varepsilon^2), \tag{76}$$

where the last step uses (75) for each fixed $\theta$.

Choose

$$\varepsilon := \sqrt{\frac{\log(2|\Theta_N|/\delta)}{2m_0}}.$$

Then $|\Theta_N| \exp(-2m_0\varepsilon^2) = |\Theta_N| \exp(-\log(2|\Theta_N|/\delta)) = \delta/2$. Therefore, with probability at least $1 - \delta/2$, for all $\theta \in \Theta_N$ with $m(\theta) \geq m_0$,

$$R(\theta) < \hat{R}_N(\theta) + \sqrt{\frac{\log(2|\Theta_N|/\delta)}{2m_0}}.$$

Let $\theta^\star$ be any solution of the empirical risk-controlled selection rule:

$$\theta^\star \in \arg\max_{\theta \in \Theta_N} \Big\{\hat{C}_N(\theta) : \hat{R}_N(\theta) \leq r, \, m(\theta) \geq m_0\Big\}, \tag{77}$$

where $\hat{C}_N(\theta) = m(\theta)/N$ is empirical coverage. On the high-probability event from above, applying (73) to $\theta^\star$ yields

$$R(\theta^\star) \leq \hat{R}_N(\theta^\star) + \sqrt{\frac{\log(2|\Theta_N|/\delta)}{2m_0}} \leq r + \sqrt{\frac{\log(2|\Theta_N|/\delta)}{2m_0}},$$

since $\hat{R}_N(\theta^\star) \leq r$ by definition (77). $\qquad\square$

# I. Algorithmic Summary

---

**Algorithm 1** Calibration and Inference for Joint Bayesian Uncertainty Fusion

---

**Inputs:** Development set $\mathcal{D}_{\text{dev}} = \{(s_i, \tilde{V}_i, y_i)\}_{i=1}^N$ (for prior & threshold only), response dimension $d$, per-response weights $\{w_{ij}\}$, hyperparameter $z_\alpha$ for final score output.

**Outputs:** Calibrated *prior* parameters $(\hat{\alpha}_0, \hat{\beta}_0, \hat{\sigma}_0^2)$, decision threshold $\hat{\theta}$.

```
# Phase 1:  Prior Calibration (Definition 4.1)
```

**for** each example $i$ **do**

    Compute statistic $s_i = g(\boldsymbol{\mu}_I, \boldsymbol{\Sigma}_I, \boldsymbol{\mu}_T, \boldsymbol{\Sigma}_T)$.

**end for**

Estimate prior parameters $(\alpha_0, \beta_0, \sigma_0^2)$ via weighted least squares, shown in Equation (66)–Equation (67).

```
# Phase 2:  Posterior Inference Evidence (Theorem 4.8)
```

**for** each query **do**

    Compute weighted scatter: $\tilde{V} = \log \det(\tilde{\boldsymbol{R}}\tilde{\boldsymbol{R}}^\top)$, $\quad \tilde{\boldsymbol{R}} = \begin{bmatrix} \sqrt{w_1}\boldsymbol{r}_1^\top \\ \cdots \\ \sqrt{w_n}\boldsymbol{r}_n^\top \end{bmatrix}$.

    Compute weight summaries $\bar{w}$ and $\nu_{\text{eff}}$.

    Compute evidence parameters (per query):

$$a = a_0(d, n) + n \log \bar{w} - \frac{n}{2}\left(\frac{n}{\nu_{\text{eff}}} - 1\right), \quad \tau^2 = \sum_{j=1}^n \psi_1\left(\frac{d+1-j}{2}\right).$$

    Compute prior mean: $m_0 = \hat{\alpha}_0 + \hat{\beta}_0\, s$.

    Compute posterior variance and mean: $\sigma_{\text{post}}^2 = \left(\frac{1}{\hat{\sigma}_0^2} + \frac{n^2}{\tau^2}\right)^{-1}$, $\quad m_{\text{post}} = \sigma_{\text{post}}^2\left(\frac{m_0}{\hat{\sigma}_0^2} + \frac{n(\tilde{V}-a)}{\tau^2}\right)$.

    Store $m_{\text{post}} + z_\alpha \sigma_{\text{post}}^2$ as the fused uncertainty score for this query.

**end for**

```
# Phase 3:  Calibration and Threshold Selection
```

Choose global threshold $\theta^*$ by risk control:

$$\theta^* = \sup_\theta \left\{ \frac{\sum_{q=1}^N \mathbf{1}\{u_q^* \leq \theta,\ \hat{y}_q \neq y_q\}}{\sum_{q=1}^N \mathbf{1}\{u_q^* \leq \theta\}} \leq r \right\}.$$

```
# Inference at Test Time
```

For a new query, compute $(s, \tilde{V}, \bar{w}, \nu_{\text{eff}})$, then $(a, \tau^2)$ analytically, then $m_{\text{post}}$.

**If** $m_{\text{post}} \leq \theta^*$ **then** output prediction; else abstain.

---

## J. More Ablations

We run more ablations in this section.

### J.1. Number of Generations

First, to examine how the number of generations affects performance across metrics, we run an ablation study on a VQAv2 test set, varying the number of sampled responses per instance from 2 to 50, following the same setup in (Lau et al., 2025). Figure 3(a) shows that AUROC generally increases as more generations are added for all methods. However, FUSE reaches strong performance with substantially fewer generations than the baselines, suggesting greater sample efficiency. This follows the same trend as in UMPIRE but FUSE slightly outperforms UMPIRE. In contrast, competing methods depend more heavily on additional samples to keep improving. Overall, these results indicate that our approach robustly extracts correctness signals even when only a small number of generations are available.

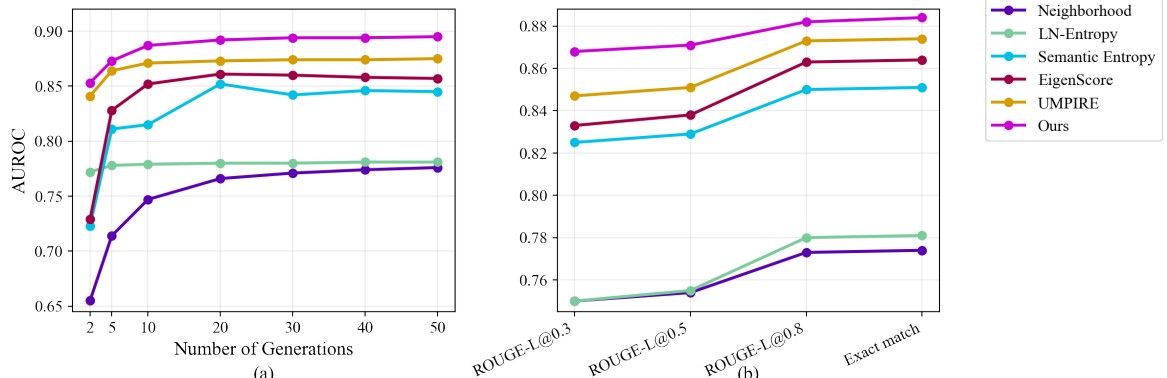

*Figure 3.* Ablations on number of generations and evaluation criteria.

| | **AUROC** (↑) | | | **Pearson Corr.** (↑) | | | **Spearman Corr.** (↑) | | | **AURAC** (↑) | | |
|---|---|---|---|---|---|---|---|---|---|---|---|---|
| **Method** | VQA | OK | AdV | VQA | OK | AdV | VQA | OK | AdV | VQA | OK | AdV |
| Prior-Only | 0.322 | 0.230 | 0.283 | 0.304 | 0.515 | 0.373 | 0.403 | 0.393 | 0.417 | 0.336 | 0.321 | 0.411 |
| Evidence-Only | 0.853 | 0.746 | 0.763 | 0.902 | 0.895 | 0.941 | 0.852 | 0.901 | 0.875 | 0.862 | 0.743 | 0.710 |
| Fused | **0.883** | **0.761** | **0.796** | **0.945** | **0.945** | **0.974** | **0.921** | **0.944** | **0.941** | **0.940** | **0.770** | **0.793** |

*Table 2.* Ablations on uncertainty sources. We report AUROC, calibration correlations (Pearson, Spearman), and AURAC (higher is better).

## J.2. Evaluation Criteria

Unlike Exact Match, which is a strict binary criterion that only counts a prediction as correct if it matches the reference answer exactly, ROUGE-L provides a graded notion of correctness by measuring the longest common subsequence overlap between the prediction and reference. This makes ROUGE-L more tolerant to paraphrases, minor wording differences, and partially correct responses—yielding a spectrum of correctness thresholds rather than an all-or-nothing judgment. Following (Kuhn et al., 2023; Lau et al., 2025), we further test our method and the baselines under multiple ROUGE-L levels. Figure 3(b) reports AUROC on a subset of the VQAv2 test set for different correctness criteria. Across all choices of evaluation function, our approach consistently surpasses the baseline methods, indicating that its advantage is not tied to any single correctness definition. Overall, the results emphasize the robustness of our method under varying criteria for judging answer correctness.

## J.3. Uncertainty Sources

We isolate the uncertainty sources and use the corresponding statistic (normalized) as the single measure of uncertainty to ablate the effect of fusing the sources of uncertainty. In Table 2, we list the performance of using prior only and evidence only. As results suggest, fusing the sources of uncertainty significantly improve the final performance in terms of all metrics. Interestingly, the evidence statistic alone is a strong baseline. This is expected as our evidence formulation is similar to UMPIRE, which is a robust baseline itself. This again emphasizes the robustness brought by fusing the uncertainty sources in a principled Bayesian approach.

## J.4. Other Ablations Rationale

**Logistic $u^\star$.** This ablation evaluates whether downstream failure can be predicted more effectively by explicitly combining the posterior mean and posterior uncertainty into a probabilistic measure directly. It also serves as a supervised calibration baseline when correctness labels are available.

**Mean-only $u^\star$.** This ablation removes the posterior dispersion term and uses $u^\star = m_{\text{post}}$ as the uncertainty score. It directly tests whether posterior variance is necessary in practice, or whether a point estimate is sufficient to capture useful uncertainty.

**Quantile threshold $\theta^\star$.** This ablation replaces any learned or tuned decision threshold with a simple global quantile rule. It tests whether the method remains usable under a lightweight, easy-to-apply thresholding strategy and avoids reliance on label-dependent threshold selection. This also helps separate the quality of the uncertainty score from the specifics of threshold tuning.

**Global tuning.** This ablation jointly tunes $(\alpha_0, \beta_0, \sigma_0^2, a, \tau^2)$ using labeled correctness indicators, rather than computing some parameters analytically. It tests whether end-to-end supervised tuning yields additional gains by absorbing dataset- or model-specific idiosyncrasies. Comparing against analytic calibration clarifies whether improvements come from the probabilistic structure itself or from label-driven parameter fitting.

**OLS fit for $\tilde{V}$.** This ablation replaces the theory-motivated likelihood calibration with a simple linear Gaussian regression between semantic evidence and the latent scale. It tests whether the distributional assumptions (e.g., Wishart-inspired moment structure) are necessary, or whether a purely empirical linear fit already captures the relevant relationship. This provides a strong "minimal modeling" baseline for likelihood estimation.

**Unweighted semantic evidence.** This ablation computes semantic evidence without weights, using the unweighted formulation from Theorem 4.4. It tests whether weighting is essential to suppress artifacts from redundant or near-duplicate samples and to better reflect semantic (rather than superficial) diversity. Any performance difference indicates the contribution of the weighting mechanism.

## K. Hyperparameters

We clarify that we select $z_\alpha$ on a held-out calibration set that is disjoint from both training and test data. Concretely, after computing the posterior mean and variance for each calibration example, we form the final score $u_i^\star = m_{\text{post},i} + z_\alpha \sigma_{\text{post},i}$ We then sweep $z_\alpha$ over a predefined grid and choose the value that gives the best performance on the calibration set under the target objective. In our case, since FUSE is used for uncertainty estimation and selective answering, $z_\alpha$ is selected to optimize AUROC on the held-out set. The selected $z_\alpha$ is then fixed and used for all test examples. In practice, $z_\alpha$ is close to 2.

## L. Qualitative Results

We demonstrate the qualitative results of FUSE on VQA datasets in Figure 4. In the 10 examples we show, the first 8 are considered "successful," where the *normalized* FUSE score correctly associates low uncertainty with incorrect answers and abstains if uncertainty is too high. We also report the NTL score, calculated as the average per-token probability across all answers. For the Motorcycle example, the sampled responses are semantically similar, thus a low FUSE uncertainty, but the NTL score is not as high as expected. For the first toilet example, both FUSE uncertainty and NTL reflect the same uncertain outcome. Similarly, for the elephant example, both metrics give the same outcome. Interestingly, for the bus energy VQA, while the VLM overwhelmingly produced "Electricity" as the response, FUSE uncertainty measure is still high, possibly due to the existence of "Gas" in the response as well as the lack of information in the input image. For the New York Yankees example, while FUSE score is low, representing a fairly certain answer, the NTL again is not as high as expected, even though New York and Yankees should be semantically close.

In the last two examples, while FUSE's uncertainty measure is extremely low, both answers are wrong. This reflects a failure mode of FUSE when the VLM is prompted to respond with numerical answers. In the first failure, the VLM confidently outputs "1940," but the aircraft shown is actually a Junkers Ju 52/3m produced in 1930. Note that in the OKVQA dataset, the answer was labeled wrong as 1945. The NTL score is also high for the "1940" answer, which is expected as the model's confidence is reflected in the per-token likelihood for this case. In the second failure, while there's a fixed answer "950," the desired behavior should be that FUSE outputs high scores like in the bus example. Interestingly, the NTL score is not as high as expected, which means the next-token likelihood does reflect the internal uncertainty the model has that FUSE fails to capture in this case. The numerical-respose results are interesting and we will probe into such cases in future work.

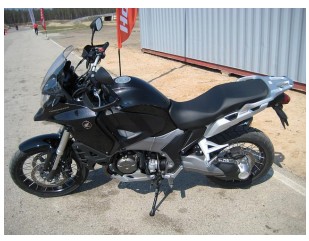

"Motorcycle racing"
"Racing"
"Race"
"Motocross"
"Motorcycle racing"

"What sport can you use this for?"

FUSE Uncertainty: 0.13
NTL: 0.49

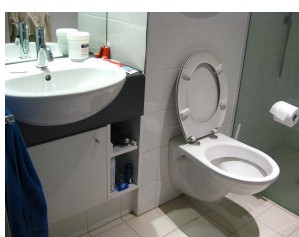

"Man"
"Vandal"
"Germs"
"Someone"
"Dog"

"Who leaves a toilet like this?"

FUSE Uncertainty: 0.91
NTL: 0.09

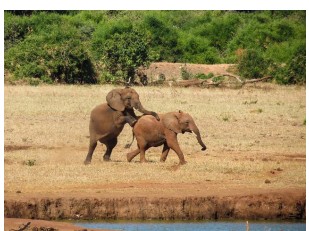

"Play"
"Play"
"Play"
"Play"
"Blend"

"Why would animals do this?"

FUSE Uncertainty: 0.22
NTL: 0.81

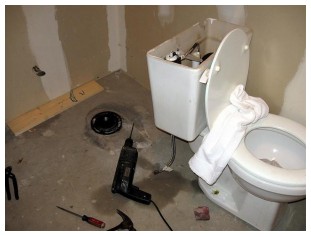

"Plumber"
"Plumber"
"Man"
"Handyman"
"Handyman"

"Who would you call if this happened?"

FUSE Uncertainty: 0.28
NTL: 0.493

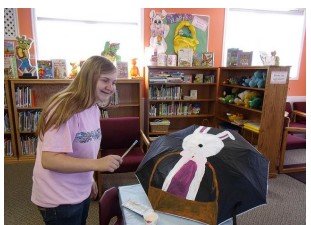
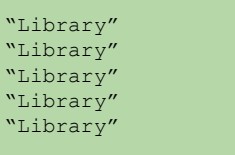

"Library"
"Library"
"Library"
"Library"
"Library"

"What area of a school might this

FUSE Uncertainty: 0.09
NTL: 0.90

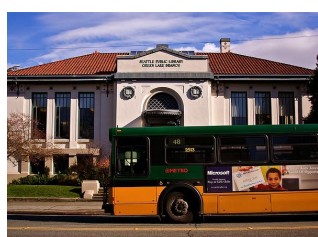

"Electric"
"Electric"
"Gas"
"Electric"
"Electricity"

"What source of energy does this vehicle use?"

FUSE Uncertainty: 0.74
NTL: 0.68

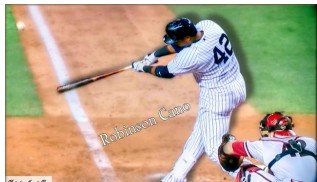

"New york yankees"
"Yankees"
"Yankees"
"New york"
"New york"

"What team is the player from?"

FUSE Uncertainty: 0.19
NTL: 0.55

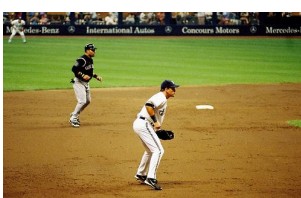

"Second"
"First"
"Catching"
"Second"
"Throwing"

"What base is this guy on?"

FUSE Uncertainty: 0.98
NTL: 0.28

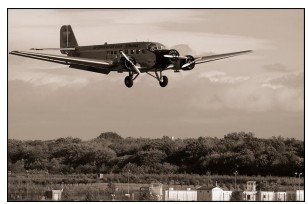

"1940"
"1940"
"1940"
"1940"
"1940"

"What year was this plane made?"

FUSE Uncertainty: 0.03
NTL: 0.79

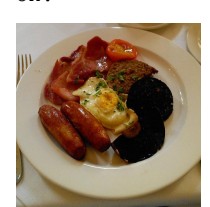
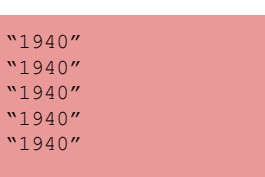
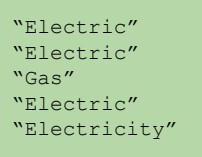

"1000"
"1000"
"1000"
"1000"
"1000"

"How many calories does this meal contain?"

FUSE Uncertainty: 0.10
NTL: 0.62

*Figure 4.* Qualitative results of running FUSE on VQA datasets. The green background represents successful cases and the red background represents failure cases.

## M. Limitations

While FUSE considers both sources of uncertainty in a principled Bayesian manner, the method, as it currently stands, mainly supports visual-language models (VLMs). Other MLLMs such as auditory models are not currently supported due to the reliance on CLIP-like encoders. However, we still think the cross-modal alignment is a good source for aleatoric uncertainty as the data representation prior, and could be generalized to other modalities as well. Moreover, the adapter for CLIP needs light retraining. We are currently looking into other alternatives. Lastly, as part of future work, we aim to deploy FUSE to robotic applications for 3D scene understanding.

