# OpenReview forum: "FUSE: Quantifying Uncertainty in Vision-Language Models by Bayesian Fusing Epistemic and Aleatoric Uncertainty"
_ICML.cc/2026/Conference — ICML 2026 regular_

### Official Review · Reviewer_6mVo · 2026-03-01

**Soundness:** 3
**Presentation:** 4
**Significance:** 3
**Originality:** 3
**Overall Recommendation:** 4
**Confidence:** 3

**Summary:**

The paper introduces FUSE, a probabilistic framework designed to comprehensively quantify uncertainty in Vision-Language Models (VLMs) by disentangling and integrating two distinct sources of uncertainty. To address the limitations of standard token-level likelihoods, the authors propose capturing both data-driven and model-driven ambiguities.  The authors evaluate FUSE on several visual question answering (VQA) benchmarks, including VQAv2, OKVQA, and AdVQA across multiple evaluation criteria, achieving state-of-the-art performance in correctness prediction (AUROC), uncertainty calibration (ECE), and selective answering (AURAC).

**Compliance With Llm Reviewing Policy:**

Affirmed.

**Final Justification:**

as mentioned in tha ack information, i think my concerns are addressed and based on my opinion and other review,  i will maintain my score. good luck

**Key Questions For Authors:**

No questions

**Limitations:**

yes

**Strengths And Weaknesses:**

1. The overall idea: combining both epistemic and aleatoric uncertainty is reliable but lack of novellty, however, The math is rigorous, featuring an elegant closed-form Bayesian update. The evaluation uses robust, industry-standard metrics like AUROC, ECE, and AURAC following prior works to prove the claims.
2. The paper tackles the critical challenge of VLM reliability and hallucination in safety-critical domains. It successfully delivers a highly effective metric for selective answering
3. Instead of relying on rigid token-level matching, the framework projects generated responses into a continuous semantic space and uses a scatter matrix to measure their dispersion.  This design elegantly accounts for semantic equivalence, perfectly fitting the generative nature of LLMs
3. The overall paper is well-written and clearly structured‘

weakness:
1. The author uses only LLaVA as the base model for evaluation following prior works, which can be accepted but evaluations on more base models are strongly accepted.
2. The authors failed to specify the value of the hyperparameter $z_{\alpha}$ and its selection strategy in the implementation details, which hinders the reproducibility of the proposed FUSE score
3. The GP adapters were trained on the VQAv2 training set, yet evaluated on the VQAv2 benchmark. The authors should clarify the impact of this potential data leakage on the test results.

---

> ### Author Rebuttal · Authors · 2026-03-24
>
> We appreciate Reviewer 6mVo for the positive assessment of the **mathematical rigor, evaluation protocol, and the usefulness** of FUSE. We are especially glad that the reviewer found the semantic-space uncertainty design well aligned with the generative nature of VLM outputs.
>
> > On novelty.
>
> We agree that combining aleatoric and epistemic uncertainty is a natural direction at a high level. Our novelty is not in the generic statement that “both matter,” but in the specific probabilistic formulation that makes them work together: (1) a GP-based probabilistic adaptation of frozen multimodal embeddings to produce an input-side prior, (2) a semantic scatter statistic over sampled response embeddings with an approximate linear-Gaussian evidence model, and (3) a **closed-form, interpretable Bayesian posterior** that fuses the two into a single calibrated statistic. Existing strong baselines such as UMPIRE and EigenScore operate primarily on generation-side semantic diversity, whereas FUSE contributes a principled, interpretable way (i.e. the monotonic relation with $n$) to incorporate encoder-side uncertainty into the same probabilistic pipeline.
>
> > On evaluation with only LLaVA.
>
> We are glad the reviewer pointed this out. Since the submission of the manuscript, we have run the pipeline on two more models CogVLM 19B and Qwen 2.5 VL 7B and measured the performance against baselines. We preview some results in the table below, which evaluates FUSE vs. baselines on the VQA-v2 dataset using AUROC.
>
> | Backbone         | LN-Entropy | Semantic Entropy | EigenScore | UMPIRE| FUSE (Ours) |
> |------------------|-----------:|-----------------:|-----------:|------------:|-----------:|
> | CogVLM 19B       | 0.43       | 0.88             | 0.87       | 0.87    |0.89    |
>  | Qwen2.5 VL 7B    | 0.67       | 0.85             | 0.84       | 0.84    |0.86    |
>
> The results show that **FUSE generalizes beyond LLaVA**, achieving the best or tied-best VQAv2 AUROC across all tested backbones. While these experiments were not part of the original submission, they provide further support that the proposed fusion framework is not tied to a single base model. We will include the full details in the revision.
>
> > On the missing hyperparameter.
>
> Thank you for catching this. We agree that the draft currently under-specifies the selected value and calibration procedure for the weighting-scale hyperparameter alpha, and we will add this explicitly in the revision. We also clarify that we select $z_\alpha$ on a **held-out calibration set** that is disjoint from both training and test data. Concretely, after computing the posterior mean and variance for each calibration example, we form the final score
> $u^\star_i = m_{{\rm post},i} + z_\alpha \sigma_{{\rm post},i}.$
> We then sweep $z_\alpha$ over a predefined grid and choose the value that gives the best performance on the calibration set under the target objective. In our case, since FUSE is used for uncertainty estimation and selective answering, $z_\alpha$ is selected to optimize AUROC on the held-out set. The selected $z_\alpha$ is then fixed and used for all test examples.
>
> > On potential data leakage from GP adapter training on VQAv2.
>
> We would like to clarify that the GP adapters are trained on the training split of the source datasets, including the VQAv2 training set, while evaluation is performed on the benchmark test split; there is no use of VQAv2 test labels or test examples during GP training. The held-out calibration set used for thresholding is also separated from test-time evaluation. So this follows the standard train/calibration/test separation rather than test leakage. We will revise the implementation details to make the split explicit.
>
> In summary, we sincerely appreciate the reviewer for their constructive feedback and acknowledgement of **the elegance and effectiveness** of FUSE. Please kindly consider raising the score if we have addressed your concerns.

---

> > ### Author Rebuttal · Reviewer_6mVo · 2026-04-02
> >
> > Thanks sincerely for your detailed reply! I think my concerns are addressed and here are some suggerstions that should not be base for AC to make final decision since it is about the writing and not the technical part.
> > 1. Though the author provide additional explanations on the design. i would like to point out that the author should emphasize the closed-form part in the main paper.
> > 2. i would like to follow reviewer tdio that such detailed paper with rigorous math should build intuitions first.
> >
> >
> > overall, i think my concern are addressed. I, however, think that rating is descent. Thanks again for your hard work and detailed response.

---

> > > ### Author Response · Authors · 2026-04-02
> > >
> > > Dear Reviewer 6mVo, thank you so much for your reply. We will add more intuition to the revised manuscript and emphasize the closed-form solution.

---

### Official Review · Reviewer_tdio · 2026-03-06

**Soundness:** 3
**Presentation:** 3
**Significance:** 3
**Originality:** 3
**Overall Recommendation:** 5
**Confidence:** 5

**Summary:**

This paper is about uncertainty estimation for multimodal large language models and vision-language models, by first estimating aleatoric uncertainty from a CLIP embedding, and epistemic uncertainty via multiple output responses from the VLM layers given variability from embedding uncertainty.

The authors propose a complex model that builds a version of CLIP with embedding uncertainty using a Gaussian process, then this uncertainty is propagated via the VLM layers that take a CLIP embedding input via latent variables to produce semantic evidence, then the aleatoric and epistemic uncertainties are fused into a single value using Bayesian data fusion, which then can be calibrated to be transformed into a probability of correctness.

The contributions are:
- A formalization of aleatoric and epistemic uncertainty in embedding-based vision-language models in order for both types of uncertainty to be quantifiable in this setup.
- A unified probabilistic model to estimate both types of uncertainties and a formulation to use bayesian methods to fuse both uncertainties to produce a single calibrated probability.
- Results on several visual-question answering datasets showing that the proposed method works better than the baselines.

**Compliance With Llm Reviewing Policy:**

Affirmed.

**Final Justification:**

The rebuttal has addressed most of my concerns/questions, and I raise my score correspondingly, my only issue that the paper needs to improve is the use of aleatoric and epistemic uncertainty, I think we agree with the authors that better terms could be used as mentioned in the rebuttal, I like input and output uncertainties, which reflect what happens more closely.

**Key Questions For Authors:**

- How is Definition 4.3 different from computing the covariance matrix of r?
- How do you validate that aleatoric and epistemic uncertainties are working as expected? In your case this can be about prompt ambiguity vs answer correctness.
- Can you explain in Defintion 4.10, how is u* a signal-to-noise ratio? It is not a ratio at all.
- The proposed method is quite complex, what are some options to reduce complexity? For example simplifying the epistemic uncertainty estimation or simply using sampling.
- Overall, why does the proposed method work better than the baselines? What is the intuition behind the improvement?

**Limitations:**

The paper is limited by its weaknesses that I have mentioned before, and the appendix contains a good general limitations statement.

The paper has a generic impact/ethics statement and I think uncertainty estimation is socially relevant, so the statement should be clear about societal impact, specially as LLMs/VLMs are being targeted as a direct application, the paper should not have a generic impact statement and the dangers of uncertainty estimation and its relationship with hallucionations should be clearly mentioned and discussed.

**Strengths And Weaknesses:**

Strengths
- Uncertainty estimation in LLMs and VLMs is an extremely important open problem, to detect and filter hallucinations and prevent end users from seeing hallucinations and making models that can say "I don't know", so at least the paper is going in the right direction.
- The formulation/definitions for aleatoric and epistemic uncertainty for embedding-based VLMs seem to be correct and useful for future research, I think its the first time I see such definitions specifically for multimodal LLMs/VLMs so they seem to be novel.
- The evaluation does show that the proposed method works better than the baselines on three VQA datasets in terms of uncertainty quality. The selection of baselines and datasets are good.


Weaknesses
- I believe the paper is hard to read and understand, at parts there is extreme level of detail, and excessive use of mathematical notation without building intuitions first, and some sections do not make sense to me. For example Definition 4.3 uses several paragraphs to basically describe the covariance matrix of r, Theorem 4.4 mentions the semantic scatter S but Equation 5 does not use S but u instead, and Definition 4.10 uses a signal-to-noise ratio that is not a SNR but just z_a standard deviations away from the mean. Overall I get the impression that this paper was written by an LLM as there are too many inconsistencies. Additionally the paper basically has 6 pages of definitions and content defining the proposed method, and then results and experiments are only in the last two pages.
- To me the fusion of aleatoric and epistemic uncertainty does not make much sense, the point of aleatoric and epistemic uncertainty is that these behave differently, and for prediction correctness, then epistemic uncertainty is important while aleatoric uncertainty should be ignored, so in a basic case fusing these uncertainties would not be a good idea. Also the bayesian formulation to use uncertainties is not motivated and described in the paper, just stated as fact. The appendix contains some derivations but first I would expect a motivation on why we should fuse those uncertainties.
- Figure 2 in the paper is misleading, as first the model estimates aleatoric uncertainty from a CLIP embedding, then uses that uncertainty to produce semantic response variability, the LLM/VLM is still deterministic so multiple answers are produced by what I assume is sampling from the VLM input uncertainty (the paper is not clear on this), so there is a connection between aleatoric and epistemic uncertainty, and then the figure shows parallel paths for aleatoric and epistemicuncertainty, when epistemic uncertainty is built on top of input aleatoric uncertainty.
- The paper lacks many important details, training of the gaussian process for aleatoric uncertainty, training of the VLM baselines, and importantly, does not evaluate the task performance, only uncertainty estimation quality.
- The evaluation is too simplistic in my opinion, it mainly evaluates AUROC, ECE, correlations, and AURAC, to evaluate the total fused uncertainty as score for prediction correctness, and this shows that the proposed method is better than baselines, but there is no individual evaluation or validation that aleatoric and epistemic uncertainties work as expected, only the fused uncertainty is evaluated, and the paper claims both types of uncertainties are disentangled but there is no proof/evidence of this.
- Ablations only make some variations of how the final statistic u is built, but I still have question about other components of the pipeline, for example how do you select a gaussian process kernel for aleatoric uncertainty, and what other forms of epistmic uncertainty could have been tested for the VLM component, and validation that the aleatoric and epistemic uncertainties are working as expected independently. For example qualitative results showing individual estimates of both types of uncertainties and relating them if answer is incorrect or question/prompt is ambiguous.
- Overall this paper makes the common mistake of showing that a complex method works better than baselines but not arguing correctly why the method works better, what are the intuitions/concepts that the community can learn and build upon, and what are the important components of the proposed pipeline. More important that a method working better on some metrics and benchmarks, is why it works better, and this paper does not address that question.

---

> ### Author Rebuttal · Authors · 2026-03-25
>
> We appreciate Reviewer tdio's feedback, especially the positive assessment that UQ for VLMs is **important**.
>
> > On presentation.
>
> We agree that the Method is detailed. However, the other two reviewers **praised the presentation**.
> -  HckD: *"...clear and explains the procedure well. It walks through the derivation process nicely, making it an enjoyable read."*
> - 6mVo: *"...well-written and clearly structured"*.
>
> We agree that we will further improve the readability in the revision.
>
> > On fusing:
>
> In VQA, correctness is not determined only by model ignorance; it also depends on whether the input itself is weakly aligned or underspecified. We motivated aleatoric uncertainty as from ambiguity in the multimodal input, while epistemic uncertainty the instability in the model’s responses due to limitations. These two sources behave differently, but that is exactly why modeling both is useful; a model can be highly confident yet still on an ambiguous input. An uncertainty estimate based only on response variability will miss an important failure mode.
>
> Bayesian fusion is motivated by the structure of the problem. We define a latent uncertainty variable u; an encoder induces a prior over u, which summarizes input ambiguity before generation. Sampled dispersion of responses provides evidence about the latent variable after generation. Once both are interpreted as information about u, Bayesian fusion is the principled way to combine them: prior captures what is uncertain from the input, and likelihood captures what the response cloud reveals additionally. This is not “just stated as fact”; the paper derives an evidence model for the semantics and then combines it with the encoder-side before the closed-form posterior. So the Bayesian step is not an arbitrary heuristic. It is the natural consequence of having a prior signal and an evidence signal that are defined over the same latent.
>
> There is also empirical support that fusion is beneficial. The appendix (Table 2) includes a source-isolation study comparing prior-only, evidence-only, and fused variants. Prior-only is weak, evidence-only is strong, and the fused model performs best overall. That is exactly the pattern one would expect if aleatoric uncertainty still contributes complementary information.
>
> > On Fig. 2
>
> We agree that the current figure can be clearer. The intended meaning is not that epistemic uncertainty is produced by propagating aleatoric uncertainty through the VLM. The figure depicts two complementary estimators: one computed from the encoder, and one computed from stochastic decoding of the VLM. The multiple answers are generated by stochastic decoding, as also reflected by sampled responses in the method and algorithm.
>
> > On details
>
> We stated that we follow Venkataramanan et al., 2025 in the method section. The current draft is compressed on implementation details, and we will expand them in the revision. The VLMs were pretrained.
>
> > On evaluation
>
> The paper’s objective is UQ for VLM outputs. Accordingly, the evaluation focuses on correctness prediction, calibration, and selective answering, which are the standard downstream criteria for uncertainty estimation. As reviewer 6mVo pointed out, we have used *"industry-standard metrics."* The method is meant to assess when the model is likely to be wrong. The AURAC metric, however, does measure the task performance.
>
> Empirically (Table 2), the prior-/evidence-only/fused comparison shows that the prior captures a distinct but weaker signal and that the evidence captures a stronger signal, with the fusion improving further. This is an empirical validation of the decomposition. We agree that adding more qualitative examples would further improve intuition, e.g., cases of prompt ambiguity versus response instability.
>
> > On complexity
>
> The method is structured in this way as it combines encoder- and generation-side uncertainty in an interpretable way. There are natural simplifications: one can use only the evidence term, unweighted scatter instead of weighted, or the final conservative score with mean. The paper already studies variations in how the final statistic is built and the source isolation.
>
> The key intuition is that token-level confidence alone is often misaligned with correctness, while generation-only dispersion methods ignore input-side ambiguity. FUSE improves by combining. The log-det construction is important as it turns dispersion into a tractable likelihood, making the fusion principled. The main takeaway is not that a complex pipeline wins on benchmarks, but that UQ for VLMs benefits from considering ambiguity in the input and instability in the response, with a fusion grounded in a closed form.
>
> > On definitions
>
> The signal-to-noise ratio was an oversight, as an earlier iteration of the work used mean/std. Def 4.3 is not merely to summarize second-order statistics, but to construct a geometric statistic whose log-det is exactly linear in u. Thm 4.4's $\Lambda$ uses $S$ (See proof).

---

> > ### Author Rebuttal · Reviewer_tdio · 2026-04-04
> >
> > I am mostly satisfied with these answers to increase my score, I think the remaining issue is that what the authors call aleatoric uncertainty might not be that, as I commented in my review, passing input through a embedding model will lead to epistemic uncertainty, so it would be best to clarify this.

---

> > > ### Author Response · Authors · 2026-04-04
> > >
> > > Dear Reviewer tdio, thank you for the follow-up. We agree this point should be clarified more carefully, and we appreciate the suggestion.
> > >
> > > Our idea was to distinguish two sources of uncertainty at different stages of the pipeline: an **input-side uncertainty signal** derived from the multimodal representation before generation, and an **output-side uncertainty signal** derived from dispersion among sampled responses after generation.
> > >
> > > We referred to the former as “aleatoric” because it is meant to capture **ambiguity already present in the image-question pair**, such as weak grounding, underspecification, or multiple plausible alignments. In the full VLM inference pipeline, the input-stage uncertainty signal is captured by this GP-wrapped CLIP embedding, so it is analogous to the notion of "aleatoric" uncertainty in classical ML. Moreover, decomposing the uncertainty signal into input-side and output-side follows naturally. However, we agree that since this signal is from a learned embedding model and GP adapter, calling it purely “aleatoric” may confuse the reader, even though it is still considered the **input-side** uncertainty in the VLM pipeline.
> > >
> > > We will therefore revise the wording to make this precise. Importantly, this clarification does not change the method itself or the role of the two components in the model. The key point is that FUSE combines a prior derived from the input representation ("aleatoric") with evidence derived from response semantics ("epistemic"), resulting in an elegant closed-form solution. We agree that the terminology should better reflect this distinction, and we will revise the paper accordingly to avoid misinterpretation of the first term.
> > >
> > >
> > > Thank you again for your thoughtful feedback, and please kindly consider raising the score if we have clarified your question.

---

### Official Review · Reviewer_HCkD · 2026-03-10

**Soundness:** 2
**Presentation:** 3
**Significance:** 2
**Originality:** 3
**Overall Recommendation:** 4
**Confidence:** 4

**Summary:**

The paper considers visual question answering and introduces an uncertainty quantification method by analyzing both the aleatoric and epistemic components. The method (FUSE) interprets aleatoric uncertainty as the ambiguity in vision-language alignment, and epistemic uncertainty as the variability in the model’s prediction. FUSE fits Gaussian process adapters, generalizing deterministic latent image and text embeddings to distributions, which then get summarized into a data representation prior, representing the aleatoric uncertainty. For epistemic uncertainty, FUSE samples responses from the VLM, and use weighted semantic scatter as the uncertainty measure. The two sources of uncertainties get fused via Bayesian inference, and is shown to outperform baselines in terms of calibration and early stopping on three VQA benchmarks.

**Compliance With Llm Reviewing Policy:**

Affirmed.

**Final Justification:**

The rebuttal has addressed my concerns on overly broad claims beyond VQA and associated assumptions, as well as more qualitative intuitions and discussions on the results.

**Key Questions For Authors:**

1. Does FUSE only work for VQA tasks with clear ground-truth answers? How would the method respond in cases where 1) the question is not directly about objects in the image, and 2) the prompt is for example “what does the image not contain?”
2. Could you analyze whether the individual aleatoric or epistemic uncertainty measure capture the sources of uncertainty meaningfully?

**Limitations:**

yes

**Strengths And Weaknesses:**

- Soundness
    - Under its set of assumptions, the paper is technically sound, with theorems and expressions sufficiently derived and proved.
    - However, there are many assumptions on the nature of the task as well as the VLM that may require more justification.
    - First, the framework is introduced as a general UQ method for VLMs. However, the method is tailored to visual question answering and would not make as much sense for other tasks. For example, in a robotics application, a VLM might be presented with a wrist camera view image with several objects on it, and be asked in which direction should the robot move. The image and textual inputs in this case might diverge, but the aleatoric uncertainty should not correspondingly increase. Since the method is only tested on VQA benchmarks, it should be made clear that it’s designed for VQA tasks.
    - Second, the aleatoric uncertainty measurement is based on a training GP adapters with a CLIP backbone, which could behave like a bag-of-words model. For example, a query asking “what’s not in the image” might lead to low data representation prior but in reality should have high ambiguity.
    - Third, the choice to represent language in latent space as Gaussians is not sufficiently backed up. The justification in Appendix G is assuming LLaVA fine-tuned with questions with a single answer, while other VLMs could more likely be fine-tuned with human preference labels, differing from the sharply concentrated distribution described in the paper.
- Presentation
    - The method section is clear and explains the procedure well. It walks through the derivation process nicely, making it an enjoyable read.
    - Figure 1 can be greatly improved. Currently, it’s unclear why the top and bottom row produce different answers, and why they should have different uncertainties. Are the two VLMs different models?
    - It would be helpful to move the algorithm forward to accompany the detailed discussion of methods, especially distinguishing between what are design choices, what needs to be trained or calibrated beforehand, and what is determined afterwards.
- Significance
    - While distinguishing between aleatoric and epistemic uncertainties is well-motivated and generally important, the paper does not analyze both sources of uncertainty in much detail. Discarding this distinction and resulting with a single uncertainty measure seem to undermine the method.
    - The complexity of the procedure and the many assumptions might limit the scope and applicability of the proposed method.
- Originality
    - The method is novel in the sense of fusing different sources of uncertainty.

---

> ### Author Rebuttal · Authors · 2026-03-24
>
> We thank the reviewer HCkD for the careful reading and for highlighting both the technical strengths and the limitations. We are glad that the **derivations and method presentation were clear.**
>
> > On scope and applicability beyond VQA.
>
> We agree that the current paper is evaluated specifically on VQA-style correctness prediction, and that this scope should be stated more explicitly. Although the introduction motivates uncertainty in broader VLM applications, the formulation and experiments in the paper are centered on multimodal queries of the form $(I,T)$ with a correctness-oriented answer event, and all experiments are conducted on VQA datasets using exact-match correctness. We will revise the paper to clarify that the current instantiation of FUSE is a VQA-focused UQ framework. It is worth noting that the broader idea of fusing input-side and generation-side uncertainty may generalize with task-specific priors.
>
> Specifically, in the robotics example, while our current aleatoric setup may not apply well (the epistemic one should still work as it measures "diversity"), our goal is not to equate aleatoric uncertainty with generic image-text divergence, but to use probabilistically adapted encoder uncertainty as an input-side ambiguity prior. Our current work on applying FUSE to **robotic navigation tasks** developed a new task-specific prior as aleatoric, and the pipeline runs as is because we do not apply the exact formulation of aleatoric uncertainty in the downstream formulation.
>
> > On the complexity and assumptions.
>
> We thank the reviewer for this important point. For complexity, our pipeline follows the standard *calibrate-then-test* pipeline in the UQ literature, and has a modest number of calibrated parameters of interest. In terms of assumptions, we applied the Gaussian assumption mainly to make the fused score tractable. In our method section, the key requirement is that response uncertainty induces a dispersion statistic with approximately monotone behavior in semantic space, which allows the log-determinant evidence to be fused with the prior in closed form. The assumption is introduced to make this relationship analytically tractable and yield the posterior in Thm 4.9, not to assert that all VLMs trained with preference supervision must satisfy the same sharply concentrated conditional distribution. We will revise the paper to clarify this scope more explicitly. Even when the true conditional response distribution is not exactly Gaussian or unimodal, our semantic evidence is computed from sampled response embeddings and **therefore functions as a summary of the empirical response cloud**. In that sense, the Gaussian model is **best interpreted as an analytically convenient latent model for this cloud**, rather than a literal claim about token-level training distributions. As shown in **our response to Reviewer 6mVo**, FUSE exhibits strong performance on **RLHF-trained models** such as Qwen 2.5 VL, suggesting that the Gaussian assumption is well grounded even on human-feedback-trained models.
>
> > On Figure 1 and the presentation.
>
> We agree with the reviewer here. The current Figure 1 is too compressed and does not make the comparison setup sufficiently explicit. We will improve the figure/caption to clarify that it is an example showing that *under stochastic sampling* of one model, next-token likelihood alone can fail to separate a hallucinated answer from a correct one. We will also move the algorithm earlier in the method section and more clearly distinguish: (i) pretrained/frozen components, (ii) GP-adapter training and calibration steps, and (iii) test-time uncertainty computation.
>
> > Finally, on isolated uncertainty sources.
>
> We apologize for the confusion we might have caused, but already includes an uncertainty-source ablation (Table 2 of Appendix J.3), comparing prior-only (aleatoric), evidence-only (epistemic), and fused uncertainty. On VQAv2, for example, AUROC goes from 0.322 with prior-only to 0.853 with evidence-only and 0.883 when fused. It is worth noting that the epistemic uncertainty alone is a fairly strong baseline as **it measures the semantic dispersion**, which is universal across all VLM tasks. While the aleatoric uncertainty alone is not particularly strong (resonating with the reviewer's point on the choice of aleatoric), when combined with the evidence, the performance increased a lot. We again argue that the prior is only set up to capture **some notion of input-side uncertainty**, which in the VQA case, is represented via disagreement. This directly evaluates whether the individual aleatoric and epistemic components capture meaningful signals, and the fused model performs best overall, supporting the complementarity claim.
>
> Again, we sincerely appreciate the constructive feedback from the reviewer and their acknowledgment of the **novelty of our method** and the **clarity of presentation**. Please kindly consider raising the score if we have addressed your concerns.

---

> > ### Author Rebuttal · Reviewer_HCkD · 2026-04-03
> >
> > Thanks for your detailed response! I appreciate the explanations, they've addressed my main concern about the scope beyond VQA. Sorry for the confusion, by analyzing sources of uncertainty I meant it would be nice to include some qualitative discussions. This can help readers better interpret the results, in addition to seeing quantitative improvements. The current qualitative examples in the appendix don't discuss separate uncertainty sources.
> > I will be happy to raise my score if this final point is addressed.

---

> > > ### Author Response · Authors · 2026-04-03
> > >
> > > Dear Reviewer HCkD, thank you for the thoughtful follow up. We appreciate the clarification, and we agree that the paper would benefit from a more explicit qualitative discussion of the separate uncertainty sources.
> > >
> > > Looking at **Figure 4** in the appendix, the examples already suggest a useful qualitative distinction between uncertainty regimes and we will expand on the intuition here.
> > >
> > > - Some low-uncertainty cases, such as the library and motorcycle examples, show sampled answers that remain tightly concentrated around a single semantic interpretation even when the wording varies slightly. These cases are consistent with low epistemic uncertainty, since the response distribution is stable, and also likely low aleatoric uncertainty, since the image-question pair is visually well grounded.
> > >
> > > - By contrast, several high-uncertainty cases show strong semantic dispersion across incompatible answers. For example, in the “Who leaves a toilet like this?” example, the sampled answers range from “Man” to “Vandal” to “Germs” to “Dog,” which are not paraphrases of one another but fundamentally different interpretations. Likewise, in the baseball-base example, the responses mix base identification with action descriptions (“Second,” “First,” “Catching,” “Throwing”), indicating that the model is not settling on a coherent interpretation. These are qualitatively consistent with high epistemic uncertainty.
> > >
> > > - Some questions in Figure 4 are intrinsically underspecified from the image alone, such as asking about motives, exact calories, or non-visible data. Examples such as “Who leaves a toilet like this?”, “Why would animals do this?”, “What year was this plane made?”, and “How many calories does this meal contain?” are qualitatively consistent with elevated aleatoric uncertainty, because the image-question pair itself does not uniquely determine a verifiable answer. In such cases, uncertainty arises not only from unstable generation but also from ambiguity already present in the input.
> > >
> > > More concretely, we will make explicit three qualitative regimes in the revision:
> > >
> > > - **Aleatoric-dominant:** the input is relatively underspecified, but sampled answers remain semantically stable. For example, a question like “Why would animals do this?” may not have a single verifiable answer from the image alone, yet the model’s samples cluster around one interpretation.
> > > - **Epistemic-dominant:** the image-question pair appears reasonably answerable, but sampled answers scatter widely across incompatible interpretations. The baseball-base example is qualitatively of this type.
> > > - **Both high:** the question is intrinsically underspecified and the sampled answers are also highly dispersed. The “Who leaves a toilet like this?” example is qualitatively consistent with this regime.
> > >
> > > We believe this will help readers better understand: *why the prior-only signal is weaker, why the evidence-only signal is already strong, and why fusion improves further.*
> > >
> > > In the revision, we will add a dedicated qualitative analysis section that reports, for representative examples, the prior-only, evidence-only, and fused scores side by side, together with the image, question, sampled responses, and correctness. We will use this to explicitly discuss cases where aleatoric uncertainty appears dominant, cases where epistemic uncertainty appears dominant, and cases where both are elevated.
> > >
> > > Thank you again for the suggestion; we agree this would make the paper substantially more interpretable. Please kindly consider raising the score if your question has been addressed.

---

### Decision · Program_Chairs · 2026-04-30

**Decision:**

Accept (regular)

**Comment:**

This paper presents FUSE, a probabilistic framework for quantifying uncertainty in MLLMs for visual question answering tasks. The method distinguishes between two uncertainty sources: aleatoric uncertainty derived from input-level vision-language ambiguity, and epistemic uncertainty estimated from semantic response diversity of sampled outputs. These are combined through a Bayesian fusion mechanism to produce a calibrated scalar uncertainty measure. The authors evaluate FUSE on three VQA benchmarks, demonstrating improvements over baselines.

The reviewers consistently acknowledge the technical rigor and soundness of the proposed framework. Several concerns have been raised in the reviews. Reviewer HCkD questioned the scope of applicability beyond VQA tasks and raised concerns about the Gaussian modeling assumptions. Reviewer tdio expressed concerns about presentation clarity, finding inconsistencies in definitions and notation, and questioned the motivation for fusing aleatoric and epistemic uncertainties rather than treating them separately. Reviewer 6mVo identified missing implementation details regarding hyperparameter selection.

The authors have provided extensive rebuttals. The authors successfully addressed these concerns with additional clarification and new experiments. They clarified that the Gaussian assumption serves analytical tractability and demonstrated strong performance on RLHF-trained models like Qwen 2.5 VL. New experimental results on CogVLM and Qwen 2.5 VL were provided to demonstrate generalization beyond LLaVA. The authors also clarified hyperparameter selection methodology and data split procedures. All the reviewers are satisfied with the rebuttal.

The paper presents a technically sound and mathematically rigorous contribution to uncertainty quantification in MLLMs. The proposed framework offers a principled fusion of complementary uncertainty sources with strong empirical results. All the reviewers' consensus supports acceptance. The authors are encouraged to revise the paper as promised in the rebuttal.